# Advanced three-dimensional X-ray imaging unravels structural development of the human thymus compartments
Savvas Savvidis[1], Roberta Ragazzini [2,3], Valeria Conde de Rafael[1,2,3], J. Ciaran Hutchinson [4], Lorenzo Massimi[1], Fabio A. Vittoria[1,10], Sara Campinoti [2,3,11], Tom Partridge[1], Olumide K. Ogunbiyi[4], Alessia Atzeni[5], Neil J. Sebire[4], Paolo De Coppi [6,7], Alberto Mittone[8,12], Alberto Bravin [8,9], Paola Bonfanti [2,3,13] ✉ & Alessandro Olivo [1,13] ✉

## Abstract

**Background** The thymus, responsible for T cell-mediated adaptive immune system, has a structural and functional complexity that is not yet fully understood. Until now, thymic anatomy has been studied using histological thin sections or confocal microscopy 3D reconstruction, necessarily for limited volumes.

**Methods** We used Phase Contrast X-Ray Computed Tomography to address the lack of whole-organ volumetric information on the microarchitecture of its structural components. We scanned 15 human thymi (9 foetal and 6 postnatal) with synchrotron radiation, and repeated scans using a conventional laboratory x-ray system. We used histology, immunofluorescence and flow cytometry to validate the x-ray findings.

**Results** Application to human thymi at pre- and post-natal stages allowed reliable tracking and quantification of the evolution of parameters such as size and distribution of Hassall's Bodies and medulla-to-cortex ratio, whose changes reflect adaptation of thymic activity. We show that Hassall's bodies can occupy 25% of the medulla volume, indicating they should be considered a third thymic compartment with possible implications on their role. Moreover, we demonstrate compatible results can be obtained with standard laboratory-based x-ray equipment, making this research tool accessible to a wider community.

**Conclusions** Our study allows overcoming the resolution and/or volumetric limitations of existing approaches for the study of thymic disfunction in congenital and acquired disorders affecting the adaptive immune system.

## Plain language summary

The thymus is the organ responsible for programming the immune system. It consists of two main compartments, named medulla and cortex. The medulla contains onion-shaped parts known as "Hassall's bodies". By imaging thymi at different stages of development with advanced x-ray methods, we gain understanding of changes that occur over time in 3D. We quantified how much of the thymus was occupied by these different components as they change with age, showing that Hassall's bodies can take up 25% of the medulla, and should therefore be considered a proper part of the thymus with a purpose. Having a better understanding of the thymus can prove important in targeting conditions such as Down syndrome and thymic tumours, as well as provide information about structure.

The thymus is a lympho-epithelial organ located in the chest behind the sternum. It is composed of two lobes developing from the third pharyngeal pouch and progressively migrating into the mediastinum during foetal development. In humans, development of thymus begins at the 5th week of gestation[1]. Between gestation weeks (GW) 10–16, the thymus differentiates into its main cellular compartments, which form the cortex and medulla[2,3]. Initially, these components are arranged as a single central medulla structure surrounded by the cortex. At GW 10, the thymic lobes start undergoing the process of lobulation i.e., sub-division into multiple lobules[2], with each lobule eventually consisting of a medulla component surrounded by the cortex. As currently reported, Hassall's bodies (HBs) (or Hassall's corpuscles) are formed by around GW 15[4–6]. These are unique to the medulla[7]; are

commonly described as "concentric bodies" composed of epithelial reticular cells filled with keratin filaments[8], and their function remains debated.

The thymus is responsible for programming the immune system[9]. Specifically, by GW 12-13 this organ starts producing T-cells (or thymic-dependent (T) lymphocytes, whose precursors originate from the bone marrow), which form part of the adaptive immune system, providing the body with cell-mediated immunity[10,11]. Matured and selected self-tolerant naïve T lymphocytes colonise the peripheral tissues[12,13].

The human thymus is characterised by an unexpected cellular complexity of its microenvironment that only emerged recently[14–17]. For instance, it is increasingly believed that the HBs play a pivotal role in the process of T-cell maturation and central tolerance[18,19]. Further studies

emphasise that ascribing a role to these structures could be essential to obtain a complete understanding of the thymic functional activity, and of its precise role in training the human immune system[20–22]. This is in contrast with their initially assumed "non-functional" status, which classified them as a "graveyard for dead thymocytes"[23]. Furthermore, the thymus undergoes an age-associated involution process with important structural and cellular changes in cortex and medulla, the mechanisms and cause of which are yet to be fully understood; it also dramatically responds to acute and chronic insults such as viral infections or cytoreductive therapy in cancer[13,24].

We identified a gap in the availability of volumetric qualitative and quantitative information on the microarchitecture of the thymic structural components (i.e., cortex, medulla and HBs) that can reflect alterations in thymic function under different circumstances. To address this gap, there have been attempts at extracting thymic morphological information to better understand the structural complexity of the thymic microenvironment[25], and at performing intra- and inter-sample assessment for the purpose of inferring biological conclusions[26–28]. However, whilst an organ's structural complexity can only be appreciated through the volumetric visualisation of its internal anatomy (ideally whilst keeping the organ intact), the majority of existing studies used destructive, 2D imaging tools, such as histology or confocal microscopy. These can inform on high-definition molecular organisation, but only on limited small tissue volumes. In addition, extracting quantitative morphometric measures from 2D images has limitations, such as the assumption that the information generated from a limited portion of the sample (e.g., the thickness of combined histological slice) is representative of the entire organ, and the variability of the extracted measures depending on the orientation of the sample upon dissection. These limitations may affect the outcome of inter-sample statistical comparisons, leading to unreliable biological conclusions. Moreover, most morphological studies are based on murine thymi which, as opposed to most mammals including humans, have typically less lobules, a regular cortex/medulla ratio and most importantly, do not have HB structures[29].

In the past, conventional CT has been employed to detect relevant macroscopic changes to diagnose thymic neoplasia and monitor overtime progression. However, the lower resolution and limits in soft tissue sensitivity of conventional CT allowed measuring the overall size, but not the anatomical compartmentalisation of the organ[30–32]. In this study we propose the use of Phase Contrast Computed Tomography (PC-CT) to extract volumetric (qualitative and quantitative) information on the human thymic anatomical components. PC-CT detects the phase shift experienced by x-rays as they traverse biological tissue, enabling the visualisation of features which are classically considered x-ray invisible[33]. PC-CT generates high-resolution images, the specific value of which depends on the used setup (e.g. 3.5 μm voxel size in the synchrotron-based part of this study) with high soft tissue contrast, and its non-destructive nature enables the visualisation of biological tissue architecture in 3D.

Most of the presented results were obtained with synchrotron-based PC-CT (SPC-CT). This was used to visualise the complex microarchitecture of the thymic anatomy, whilst enabling the distinction of all its anatomical components. For each analysed sample, the medulla, cortex and HBs were quantified in a volumetric manner.

By combining the information extracted from all imaged thymi, we obtain what is, to the best of our knowledge, a novel perspective on the structural and functional changes the thymus undergoes during this period of maturation. Specifically, we assess 1) the process of differentiation of the thymus into the medulla and cortex and subsequent changes in their relative organisation, and 2) the HBs formation and growth with age. Of note, the visual and quantitative information extracted from the SPC-CT datasets is validated against standard histology, immunofluorescence and complemented with a flow-cytometry dataset. Finally, we generate tomographic slices, namely hybrid computed tomography (H-CT), using a laboratory-based edge illumination (EI) system that show comparable quality to those obtained with SPC-CT. This demonstrates the potential of using a highly accessible laboratory system to tackle structural complexity of tissues such as the thymus in health and disease as well as for the guidance of thymic tissue engineering protocols.

## Materials and methods

### Ethics and sample cohort information
Postnatal thymi were donated by patients undergoing cardiothoracic surgery at the Great Ormond Street Hospital. Written informed consent was obtained from the patients or legally authorised representatives under ethical approval (REC No 15/YH/0334 and 07/Q0508/43-06-MI-13). Human foetal thymi were provided by the Joint MRC/Wellcome Trust Human Developmental Biology Resource (HDBR) under informed ethical consent with Research Tissue Bank ethical approval (REC No 08/H0712/34+5 and 08/H0906/21+5).

The cohorts of samples used for PC-CT imaging and FACS analysis were age matched. Specifically, 15 human thymi were considered for the PC-CT imaging, including 9 foetal (age range CS23 and 10–22 GW) and 6 postnatal (age range 19 days old to 12 MPN), whilst the FACS analysis was performed using 18 thymi, 9 foetal (age range 12–21 GW) and 9 postnatal (age range 1–11 MPN).

### Pre-imaging sample preparation
Prior to imaging, all perinatal samples underwent a 24 h fixation in 10% buffered formalin or 4% paraformaldehyde solutions whilst stored at 4 °C, followed by dehydration in graded ethanol-water cycles up to 100% ethanol.

Human thymic tissues used for PC-CT imaging were further critically point dried (CPD) using $CO_2$ (as per Savvidis et al.[34]). While in principle imaging hydrated samples is possible, we felt this was a safer preparation protocol in view of the need to transport the specimens to an overseas synchrotron and back.

### Histological slices
Following PC-CT imaging, the samples were rehydrated in 10% buffered formalin solution for a 24 h period whilst stored at 4 °C; then dehydrated in alcohol in preparation for paraffin embedding.

All samples used for histology were embedded in paraffin after ethanol dehydration sliced to 4–5 μm thickness using a microtome, and baked for 60 min at 60 °C. The slides were stained with Haematoxylin and Eosin (H&E).

### Immunofluorescence
Fixed, dehydrated and paraffin embedded human thymic sections were sliced to 4 um thickness and baked as per above. Dewaxing was carried (E2D, Tissue-Tek Prisma), followed by heat-induced antigen retrieval via microwave oven incubation of the slides in citrate buffer pH 6.00. Samples were incubated at room temperature for 2 h in a permeabilising and blocking solution (0.5% v/v Triton X100 in 5% Normal Donkey Serum (NDS)) and next incubated over night at 4 °C with anti-KRT10 mouse antibody (1:200; Santa-Cruz SC-53252). Slides were then stained with Hoechst for nuclear detection (Sigma, 1:500) and secondary antibody anti-mouse AF647 (1:500, JacksonImmuno, 715-605-150). Antibody solutions were prepared in PBS with 0.01% Triton-X100, 5% NDS and 3 washes 5 min each in PBS were carried out after each incubation.

Slides were mounted and imaged using the Zeiss 710 upright microscope, with Zen2.3sp1 software, 20x objective lens. 2D images were acquired and analysed using ImageJ version 2.9.0 and QuPath version 0.4.3.

### Flow cytometry analysis
Thymic tissues were dissociated to single cell suspension with enzymatic treatment (0.4 mg/mL Collagenase D (Roche), 0.6 mg/mL Dispase II (Gibco), 40 mg/mL DNAse I (Roche)) for around 30–45 min, using the Gentle MACS machine (Miltenyi). After the dissociation, the supernatant was collected, passed through a cell-strainer (100 μm), centrifuged at 1200 r.p.m. for 5 min and counted with trypan blue (SIGMA-ALDRICH) to assess viability. For postnatal samples, cell suspension was depleted for $CD45^+$ and $CD235ab^+$ cells by staining them with biotinylated antibodies,

**Table 1 | Antibodies used for the FACS analysis**

| CD235ab (glycophorin A and B) | Biotin | 1:100 | HIR2 | Human | BioLegend | 306618 |
|---|---|---|---|---|---|---|
| CD45 | Biotin | 1:100 | a30-F11 | Human/Mouse | BioLegend | 103104 |
| CD45 | APC | 1:200 | HI30 | Human/Mouse | BioLegend | 304011 |
| CD205 | PE | 1:400 | HD30 | Human | BioLegend | 342203 |
| EpCAM (CD326) | PE-Cy7 | 1:100 | 9C4 | Human | BioLegend | 324222 |
| Zombie Aqua™ Fixable Viability Kit | | 1:500 | | | BioLegend | 423102 |

then incubating with magnetic negative beads (Magnisort SAV Negative Beads, Invitrogen) and placing the suspension into a magnet (STEMCELL Technologies) for 10 min. The flowthrough fraction was collected, and the enriched fraction CD235ab⁻CD45⁻ was stained for surface markers to analyse epithelial cells. Foetal tissues below GW17 stage were not enriched because of small cell numbers and potential cell loss. Single-cell suspensions were stained with *ad hoc* antibody mix in Hanks Balanced Salt Solution (HBSS, Life Technologies) supplemented with 2% FBS (Life Technologies) for 30 min on ice. DAPI (SIGMA-ALDRICH) or Zombie Live-Dead dye (Invitrogen) was used to discriminate live from dead cells. FACS phenotypic analysis was performed using Fortessa X-20 machine (BD Bioscience) and FlowJo™ software. Antibodies used for this analysis presented in Table 1.

Examples of the FACS analysis for a foetal and a postnatal thymus are presented in Fig. 3b.

## Phase contrast computed tomography (PC-CT) systems

**PC-CT system 1: Synchrotron Free Space Propagation (FSP)**. The majority of the tomographic slices shown in this study were obtained with a synchrotron-based FSP system at the ID 17 biomedical beamline of the ESRF. The X-ray source had a horizontal and vertical full width at half maximum (FWHM) of 123 µm and 24 µm, respectively. The radiation originating from a wiggler was monochromatised using a bent Laue/Laue silicon (1,1,1) crystal to an energy of 35 keV. The sample stage was placed at around 150 m from the source, and the sample-to-detector distance was of approximately 3.45 m. Projections were acquired at each angular step using a PCO 5.5 camera coupled to a 350-µm thick Ce:YAG scintillator via a 1.75x lens system[35] yielding an effective pixel size of 3.7 µm, which is slightly demagnified at the sample by the moderate beam divergence introduced by the bent Laue monochromator; the effective pixel size at the sample was measured experimentally and found to be of approximately 3.5 µm. It should be born in mind that the effective resolution is usually 2-3 times larger than then effective voxel size. The specific scanning parameters varied depending on the physical size of the thymi. For samples small enough to fit within the detector's field of view, 2000 equally spaced projections were acquired through a 180° sample rotation. The exposure time per projection was 0.2 s, resulting in a total exposure time of approximately 400 s. For samples larger than the detector's horizontal field of view, 4000 equally spaced projections were acquired through a 360° sample rotation with the axis of rotation at the edge of the field-of-view, and projections between 180° and 360° "flipped" and joined to the 0°−180° ones to create a full dataset over 180°. Also in this case the exposure time per projection was of 0.2 s, resulting in a total exposure time of approximately 800 s. For samples larger than the detector's vertical field of view, multiple scans were acquired at different vertical displacements of the sample, and the resulting volumes stitched together. Tomographic slices generated using this system are referred to as Synchrotron Phase Contrast Computed Tomography (SPC-CT) slices.

**PC-CT system 2: Laboratory-based Edge Illumination (EI)**. For assessing the compatibility between lab and synchrotron systems, one of the laboratory-based EI systems available at UCL was used to scan a postnatal thymus that was initially imaged at the ESRF. The setup comprises of a rotating anode x-ray source, a 2D, "area" x-ray detector and two apertured masks (the relative alignment of which enables the detection of x-ray phase shifts), one located upstream of the sample and the other in contact with the detector. Specifically, the source is a Rigaku MicroMax 007 x-ray tube with a molybdenum target, generating a focal spot with horizontal FWHM of 70 µm. It was operated at 40 kV and 20 mA, outputting a broad polychromatic spectrum with a mean energy of approximately 21 keV. The detector is a Hamamatsu C9732DK flat panel CMOS image sensor with a $120 \times 120$ mm² sensitive area subdivided in $2400 \times 2400$ square pixels 50 µm in side, with an effective pixel size of 12.5 µm, defined by the design of the apertured masks. The masks were fabricated by Microworks GmbH (Karlsruhe, Germany) by electroplating a 300-µm-thick layer of gold on a 1-mm thick, patterned graphite substrate. The EI system components and physical principles are described in Savvidis et al.[34].

This EI system can be used to perform various acquisitions for generating phase-base images, such as "phase-shift computed tomography"[34],"hybrid computed tomography"[36] and "beam-tracking"[37]. For the purpose of this work, we used the "hybrid computed tomography" (H-CT) approach, as its underlying principles are the most similar to the acquisition and analysis methods used at the ESRF. 1200 equally spaced projections were acquired through a 360° sample rotation. The exposure time per projection was 1.2 s resulting in a total exposure time of 3.2 h. Tomographic slices generated using the EI system are referred to as H-CT slices.

## Phase retrieval and tomographic reconstruction

The acquired projection images consist of a combination of phase and attenuation effects[33]. For extracting the phase signal, the projections were processed using an algorithmic implementation of the method developed by Paganin et al.[38], and by adaptively adjusting the δ/β parameter until the desired level of image quality was obtained. The phase-retrieved projections were fed to a tomographic reconstruction routine based on standard filtered backprojections. For the SPC-CT slices, phase retrieval and reconstruction were performed using ESRF in-house software PyHST2[39]. For the H-CT slices obtained in the UCL lab, phase retrieval and reconstruction were performed in MATLAB (ver. R2022b) using a dedicated script (as per Diemoz et al.[36]) and the ASTRA toolbox[40–42], respectively.

## Data visualisation tool

2D and 3D visualisation was performed using the open source image processing packages Fiji[43] and Drishti[44–46], respectively. Fiji also features the WEKA plug-in used to automatically segment the H-CT images in Supplementary Fig. S6.

## Statistical testing

Statistical testing was performed using the Wilcoxon rank sum test (two-sided) in MATLAB (ver. 2022b). Statistical significance was considered for $p$-value $\leq 0.05$. Sample populations are described through their mean and standard deviation (mean ± STD), and for each group the population number (n) is reported in the corresponding figure in the manuscript.

## Quantitative measures

**PC-CT based volumetric medulla content.** for each sample, we selected 300 consecutive slices. This was the maximum number of slices available in the smallest samples, and we opted to use the same number of slices for all samples for consistency, by selecting the central 300 slices in

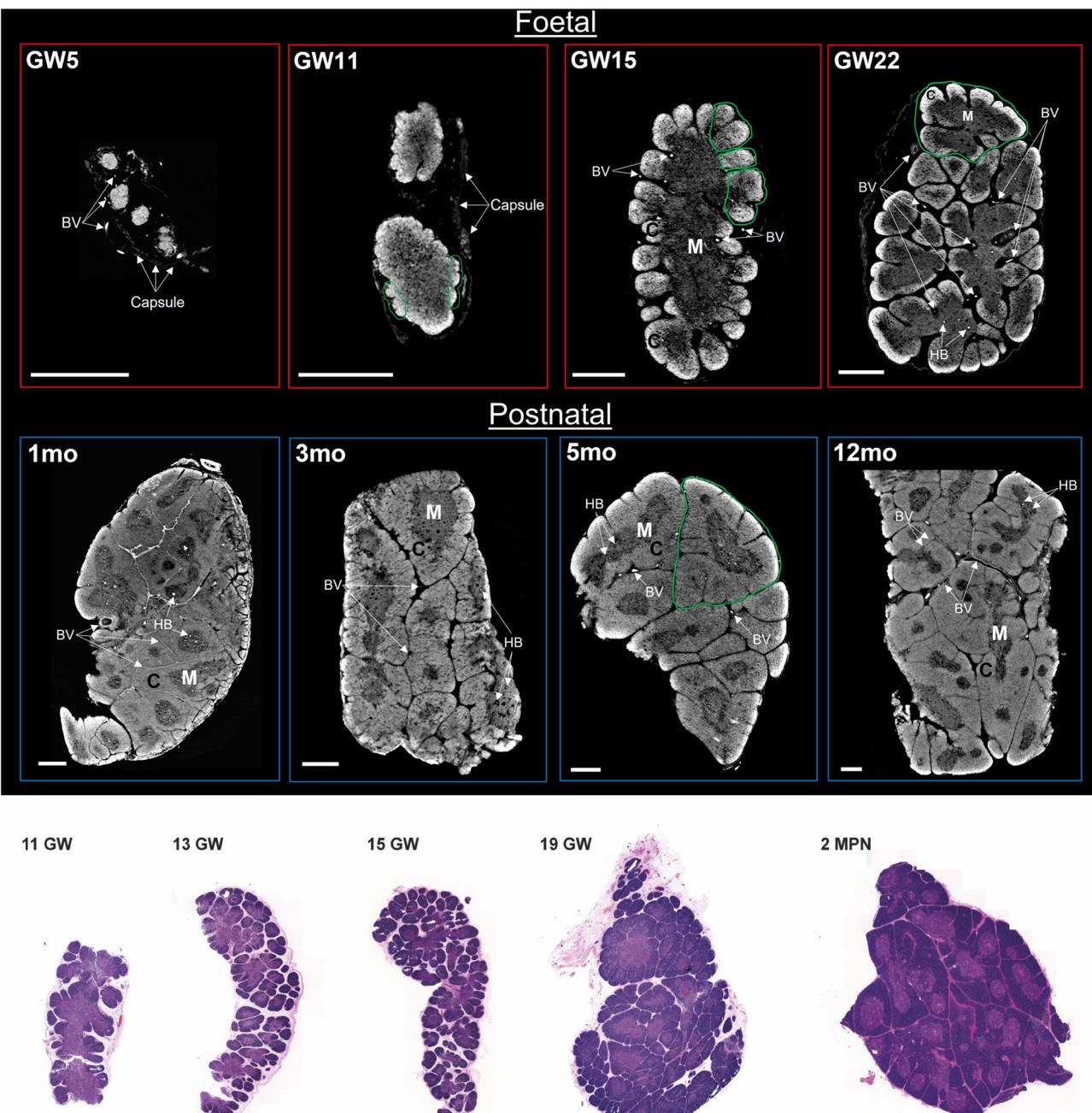

**Fig. 1 | Mapping of age associated structural changes of the thymic anatomical components via SPC-CT and histology. a** SPC-CT slices highlight the structural changes of the human thymus during maturation, starting from the foetal period to the early postnatal stages. The age of each sample in Carnegie Stage (CS), Gestational Week (GW) and month post-natal (MPN) is provided in the top left corner of each panel. Examples of areas undergoing lobulation (GW11 and GW15) and fully-formed lobules (GW22 and 5 MPN) are circled in green. The start of thymic corticomedullary definition is visible at GW11, and more evident at GW15, with compartmental organisation as a single central medulla and an outer cortex. Such conformation is repeated within each lobule from GW22 onwards. **b** Representative images of H&E staining on foetal and post-natal thymi supporting similar organogenesis as observed by SPC-CT. C cortex, M medulla, BVs blood vessels, HB Hassall bodies. Scale bars: 500 μm.

larger samples. On these 300 slices, we used a Machine Learning (ML) based segmentation algorithm (https://github.com/aleatzeni/SmartInterpol)[47] to generate binary masks representing the cortex and medulla structures for each. The same procedure described in the above reference was used to train the algorithm, consisting in manually segmenting a sub-set of the slices at regular intervals throughout the volume. The extracted cortex and medulla binary masks were used for obtaining the volumetric medulla content for each sample. This was calculated as the volumetric percentage of the medulla over the entire thymus (medulla and cortex combined), according to Supplementary Eq. (S1.1).

The medulla percentage value is a normalised quantitative measure, representing the content of the core thymic structures (cortex and medulla) of each sample, which we used to assess changes in these structures as a function of age. Specifically the samples were divided into 2 groups, foetal and postnatal, and an inter-sample statistical comparison was performed based on statistical testing (see above), with the results reported in Fig. 3a. This measure provides a volumetric normalised alternative to previous approaches applied to rat-derived thymi[26], where the corticomedullary ratio was extracted through manual measurements performed on histological slices, with a single histological slice assumed to represent an entire sample.

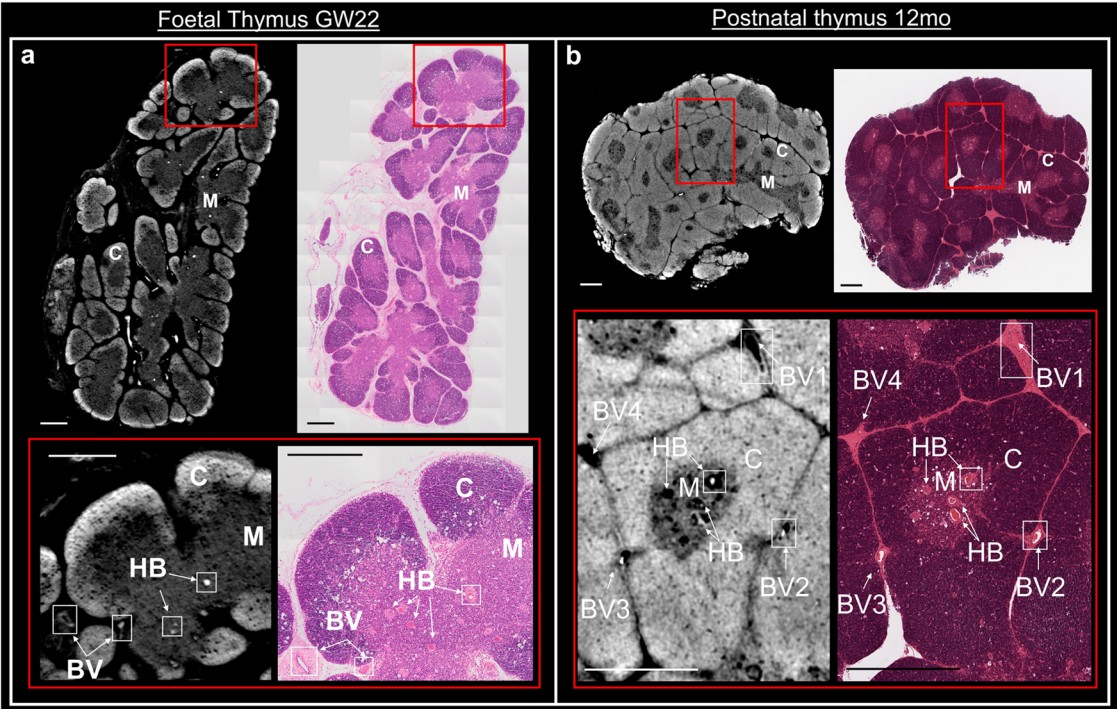

**Fig. 2 | Intrasample comparison between SPC-CT and corresponding histological slices.** Reconstructed SPC-CT slices (left) and H&E staining (right) of **a** GW22 foetal thymus & **b** a 12 MPN thymus. The inset (red square) highlights an area of interest in each image, shown below, depicting typical anatomical aspects of the samples at their respective stage with exact correlation between the two imaging tools. C cortex, M medulla, BV blood vessel, HB Hassall bodies. Scale bars: 500 μm.

To show that our choice of using the central 300 slices does not affect the final results, we have repeated the above analysis on the entire volume for a foetal (900 slices) and a postnatal (1200 slices) thymus, and compared the extracted results to those obtained from the central 300 slices. The results, presented in supplementary Fig. 3 and Supplementary Table 2, demonstrate that practically identical outcomes are obtained in the two cases.

**Volumetric HBs content**. The same 300-slice volume used for the volumetric medulla content calculation (Supplementary Eq. (S1.1)) was used to extract a quantitative measure of the volume occupied by the HBs. The HBs were manually segmented for 8 samples, 2 foetal and 6 postnatal; no HBs were detected in the remaining 5 foetal samples. Manual segmentation was performed by SS under PB's supervision; it was based on morphology, and both dark and bright (where present) regions were included. By using the manually segmented HBs binary masks in combination with the corresponding ML generated medulla masks, we calculated a volumetric percentage of the HBs over the medulla for each sample based on Supplementary Eq. (S1.2).

This value represents a normalised volumetric measure of the HBs content of each sample, which was used to assess changes in the thymic HBs content with age. This measure provides a volumetric normalised alternative to previous methods based on histology[27], which again assumed a single histological slice to be representative of the HBs content of the entire organ.

**Histology-based medullary area content**. The relative area occupied by Medulla was quantified on H&E slices of perinatal samples, by training Random Trees pixel classification on QuPath. Relative medulla area (in %) was calculated based on Supplementary Eq. (S1.3).

Samples were divided into 4 groups: 3 foetal, based on the developmental stage, and 1 neonatal. inter-sample statistical comparison was performed based on statistical testing (methods), with the results reported in Fig. 4b.

**FACS-based medulla content**. The medullary (EpCAM$^{pos}$CD205$^{neg}$) and cortical (CD205$^{pos}$EpCAM$^{low}$) cell percentages extracted for each thymus through the FACS analysis were used for generating the FACS-based medulla content calculated as per Supplementary Eq. (S1.4).

The samples were divided into 2 groups, foetal and postnatal, and an inter-sample statistical comparison was performed based on statistical testing (methods), with the results reported in Fig. 4e.

### Reporting summary
Further information on research design is available in the Nature Portfolio Reporting Summary linked to this article.

## Results
### SPC-CT digital slices reproduce thymic structural information provided by histological thin sections
To study how human thymus morphogenesis occurs, we performed SPC-CT at the European Synchrotron Radiation Facility (ESRF), for the imaging of critically point-dried foetal ($n = 9$) and neonatal ($n = 6$) human thymi, ranging from Carnegie Stage 23 (CS23) through Gestational Weeks (GW), to 12 months postnatal (MPN). SPC-CT slices allowed identifying thymic core structural components, namely cortex, medulla, and HBs, at different developmental stages (Fig. 1a).

The age range considered in Fig. 1a enabled us to visualise the still immature thymus at CS23, followed by the division into lobules (lobulation) and progressive compartmentalisation into cortex and medulla, both expected to be starting at GW10 as per previous reports[2,3]. Analysis of thymus at GW11 demonstrates that the process of lobulation and compartmentalisation has begun: lobule formation is observed externally (starting from the outermost parts), while cortical and medullary definition is still limited. At GW15, there is further progression of lobulation towards the centre of the organ, and cortico-medullary architecture can be easily distinguished. At this stage, the medulla is a single structure located at the centre of the organ and surrounded by the outer cortex. These structural features change rapidly as the process of compartmentalisation and division into lobules reaches completion at GW22, where subdivision into lobules has advanced throughout the sample, creating distinct individual units, each

**Fig. 3 | Volumetric visualisation and virtual dissection of the internal anatomy of the human thymus.** 3D rendering of whole thymi (left images) and virtual dissection across two planes (centre & right images) at **a**. GW15, **b** GW22, and **c** 12MPN. Age-associated anatomical changes are evident from **a** to **c**; with lobule formation (indicated by green contours) starting from the external part of the organ (**a**), progressing throughout the organ (**b**) and eventually leading to a reduction in inter-lobular space (**c**). Corticomedullary organisation is observed as a single central medulla and outer cortex in foetal thymus GW15 (**a**), which is replicated in the individual lobules of foetal GW22 (**b**) and postnatal 12MPN (**c**). The morphology of the HBs is visualised volumetrically in **b** and **c**; a more detailed visualisation of their shape is provided in Fig. 6. C cortex, M medulla, HB Hassall's body, BV Blood vessel. Scale bars: 500 μm.

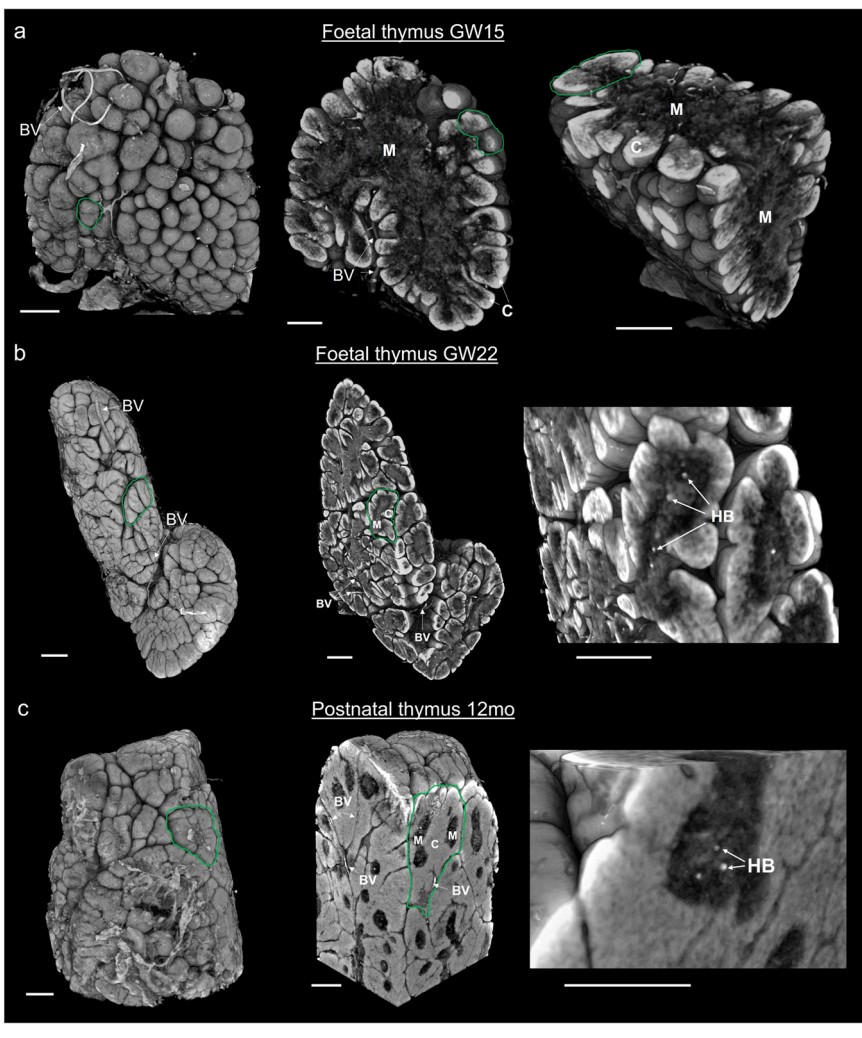

consisting of a central medulla surrounded by an outer cortex. Furthermore, we note that the division into lobules is accompanied by the formation of gaps between them. All four postnatal samples (1, 3, 5 and 12 MPN, Fig. 1a) demonstrate a compact arrangement of their lobules, with the inter-lobular spaces highly reduced in size. Examples of HBs are labelled in the medulla of samples GW22 and 1-12 MPN in Fig. 1a and in GW19 in Supplementary Fig. S1.

The structural changes observed by SPC-CT slices were comparable to those appreciated by conventional histological analyses, such as haematoxylin and eosin (H&E). A representative histological series highlights lobulation and corticomedullary differentiation progressing from early foetal stages to the third trimester, where lobes are fully formed, and interlobular spaces become more compact as seen in the 2 MPN sample (Fig. 1b). Note that these examples come from different specimens and are simply meant to show that a similar evolution is observed with SPC-CT and more conventional approaches, with a more specific, direct comparison between the SPC-CT and H&E being presented below, where the same samples were sectioned and imaged using H&E after SPC-CT scans to further validate the digital slicing with standard histological protocols. Manual orientation of physical microtome sectioning was matched with digital slicing. A clear correlation is observed between SPC-CT and corresponding histological slices for both the foetal and the postnatal thymi, as shown in Fig. 2. Every expected structural component (i.e., Cortex (C), Medulla (M), Hassall's bodies (HBs) as well as blood vessels (BVs)) is visualised in the SPC-CT slices, with their identification confirmed by the corresponding H&E section. The difference lies on the image acquisition type: H&E relies on staining, and thus dense structures appear darker.

In PC-CT, contrast depends directly on tissue density, and denser areas appear brighter. Therefore, while appearance of the sample differs, similar visual analysis can be achieved through H&E and PC-CT slices, hereby validating the latter as a reliable approach.

The visual comparison of Fig. 2 is extended to quantitative values by adapting the volumetric medulla content estimation to 2D slices (see Methods), with the results presented in Supplementary Fig. S2. The extracted quantitative values are shown in Supplementary Table 1, demonstrating a good quantitative agreement.

## Volumetric rendering provides a quantitative depiction of thymus morphogenesis

In addition to 2D visual compatibility with histology, the 3D nature of SPC-CT enables the visualisation of structural components and distribution across all anatomical planes. For example, volumetric visualisation of the GW15 sample (previously presented in Fig. 1a) enabled to carry virtual dissection of the organ across different axes, thereby confirming the early stage of lobulation (Fig. 3a). Furthermore, the cortico-medullary organisation (single central medulla – outer cortex) is consistent through the entire organ. The inter-lobular spaces at GW22 are also better appreciated as volumes, and homogeneously become more compact across the entire 12 MPN thymus (Fig. 3b, c).

Beyond visualisation, the possibility to quantify volumetric changes independently for each organ compartment represents a substantial advance to track developmental and functional changes. In the case of thymus, changes in medullary-to-cortical ratios are important, as these reflect thymopoietic activity over the course of life. Previously, conventional

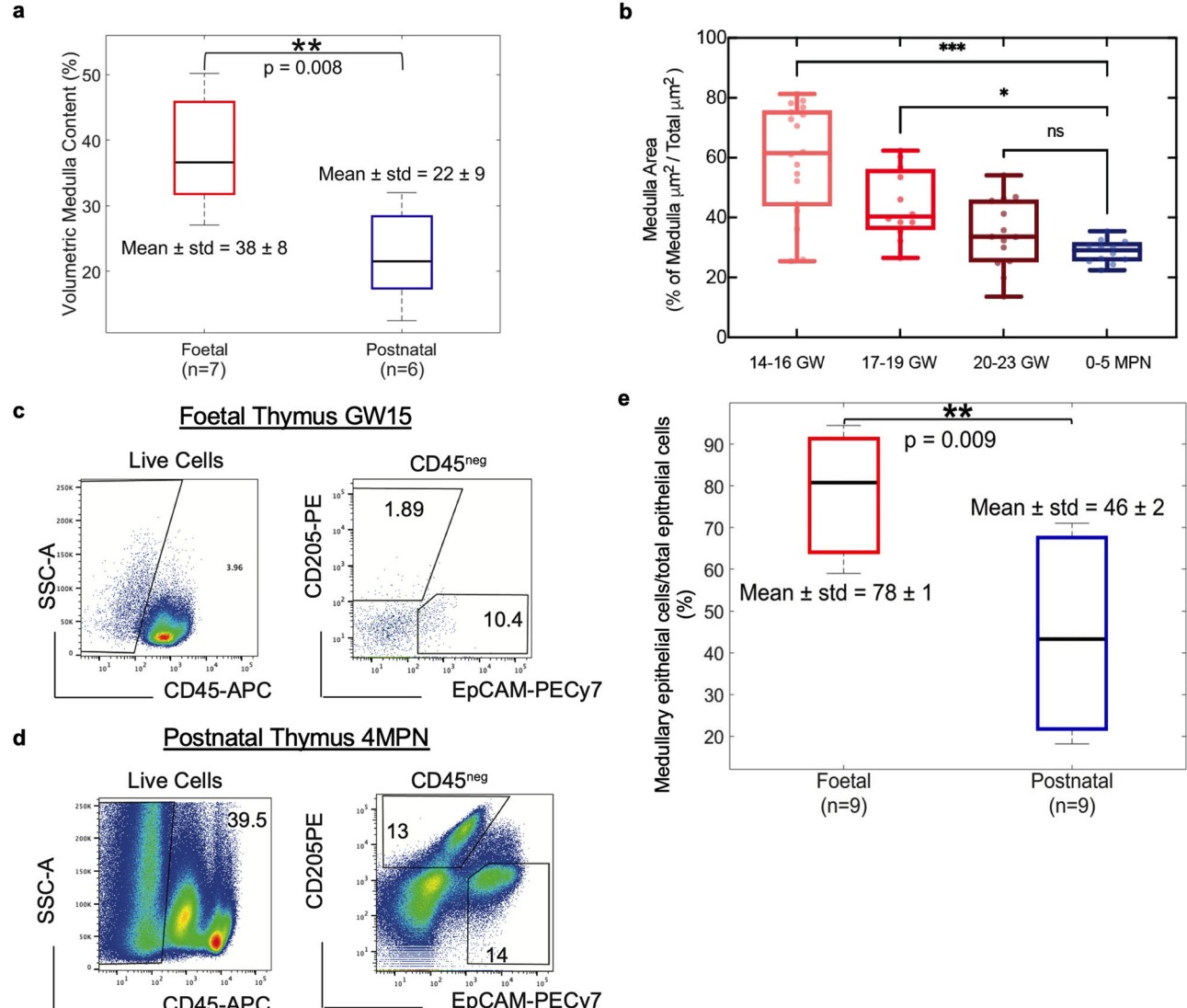

**Fig. 4 | Quantitative measures report age associated changes in thymic compartments. a** Volumetric quantification of medullary content in foetal ($n = 7$) and postnatal ($n = 6$) SPC-CT datasets. The foetal thymi demonstrated a significantly higher medulla content ($38 \pm 8\%$) than the postnatal thymi ($22 \pm 9\%$). 2 out of 9 foetal thymi (ages <GW11) were yet to start specialising into medulla and cortex compartments and were therefore excluded from this analysis. **b** Machine learning-based quantification of relative medullary area across development in H&E slices ($n = 5$ per group, with $n = 3$ technical triplicates). Samples were grouped into 4 categories, according to the developmental stage. Representative flow cytometry analysis of **c** foetal (GW15), and **d** postnatal (4 MPN) thymi. The percentage of EpCAM$^{pos}$CD205$^{neg}$ and CD205$^{pos}$EpCAM$^{low}$ was calculated on total CD45 negative fraction, quantifying medullary and cortical epithelial cell percentages, respectively. The postnatal sample was negatively enriched for CD45 as previously described[15,17]. **e** Flow-cytometry based comparison of medullary epithelial cells content (EpCAM$^{pos}$CD205$^{neg}$) relative to total epithelial cell counts. Results demonstrate a significantly higher medullary EC content in foetal thymi ($n = 9$, $78 \pm 1$) when compared to the post-natal counterparts ($n = 9$, $46 \pm 2$). p-values: <0.001(***); <0.005(**); >0.05(ns).

CT was employed to volumetrically quantify the total volume of murine thymic lobes, and match resulting values with histological projections. However sub-compartmental characterisation was not achieved[48]. Therefore, we set out to quantify volumetric changes in medullary content relative to cortex, across perinatal stages by measuring 300 SPC-CT slices per sample (Supplementary Fig. S3). A comparison between foetal and postnatal medullary contents is provided in Fig.4a. Mean relative medullary volume in foetal samples ($38 \pm 8\%$) was significantly higher than the postnatal counterpart ($22 \pm 9\%$) (**$p = 8 \times 10^{-3}$**). To further dissect these changes within developmental stages, calculation of medulla/cortex ratio was obtained through Random Forests clustering of a series of histological images ($n = 5$ biological replicates per group; $n = 3$ technical replicates). As shown in Fig. 4b, a gradual decrease of the total area occupied by the medulla is observed: from $60 \pm 18\%$ at GW14-16, to $44 \pm 12\%$ for GW17-19, and $35 \pm 11\%$ in the GW20-23 range. However, no significant difference in the relative medullary content was found between GW20-23 stages and 0-5MPN samples. These findings suggest that from approximately GW21, the thymus assumes the structure found in fully formed postnatal samples, and little conformational changes happen between these stages. It should be noted that comparisons of this type are limited to single (or series of) slices due to the 2D nature of histology, which is why high variability in medulla/cortex ratios was observed within foetal sample groups in Fig. 4b (well-spread boxplots). Since PC-CT allows extracting corresponding quantitative values for entire volumes, it should be used to further uncover perinatal structural changes by studying more samples. Nonetheless, our data clearly outline a decrease in medullary volume, occupying >30% of the neonatal thymus, which could reflect observed decrease in naïve T-cells in post-natal stages[49]. This information highlights the important structural changes that this organ undergoes during early life programming of the thymus.

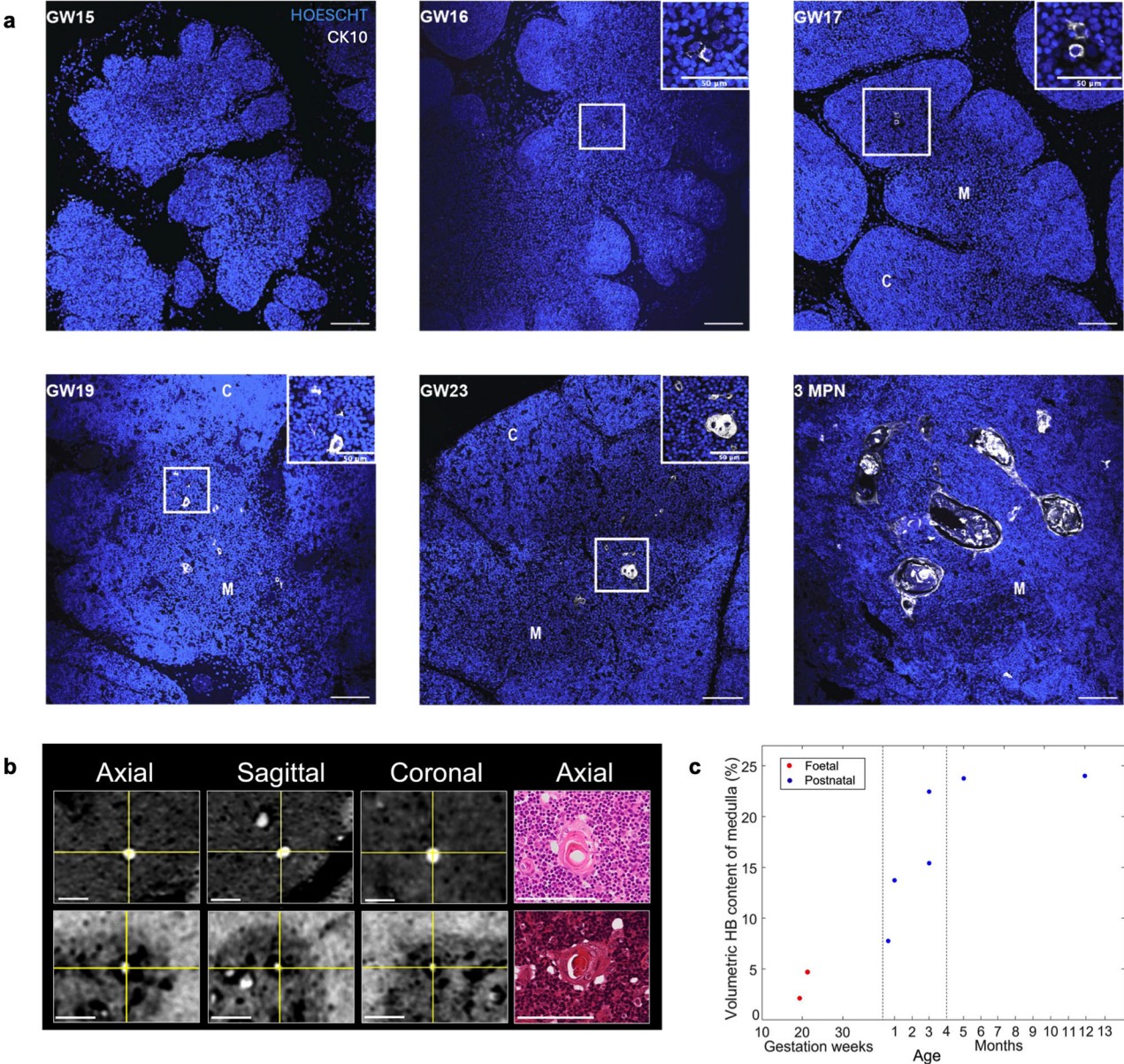

**Fig. 5 | Qualitative and volumetric quantification of HBs across development.**
**a** Representative immunofluorescence images ($n = 10$) showing appearance of
KRT10 positive cells (grey) from GW16 with progressive development of HBs from
GW19. Scale bars: 100 µm, nuclei counterstained with Hoescht. **b** Orthogonal views
(axial, sagittal and coronal) of one of the HBs found in samples depicted in Fig. 2a
and b, respectively. The corresponding histological slice (only axial plane available)
is also shown. **c** SPC-CT based Volumetric HB content of the medulla as a function
of developmental stage from foetal ($n = 2$) and postnatal ($n = 6$) thymi. Vertical lines
subdivide the graph in three age ranges where the observed behaviour differs sub-
stantially. Scale bars: 100 µm.

Volumetric measurements of cortical and medullary compartments
provide an accurate way to compare samples across developmental trajec-
tory. Most of cellularity in the postnatal thymus is represented by developing
thymocytes while stromal/epithelial cells are less than 0.5–2%[14–17]. Epithelial
cells guide the maturation of developing T cells across the organ and are
subjected to changes while the organ develops[50]. To understand whether the
epithelial cell compartment contributes to volumetric changes in cortex and
medulla during development, we performed flow-cytometry analysis on
total dissociated thymic samples (Fig. 4; Supplementary Fig. S4). We studied
stromal (CD45neg) population and analysed expression of cortical
(EPCAM^{low}CD205^{pos}) and medullary (EpCAM^{pos}CD205^{neg}) epithelial
populations to determine their abundance. We observed that foetal ($n = 9$)
have significantly higher medullary epithelial cell content ($78 \pm 1$) than the
postnatal ones, thereby reflecting relative quantifications of the anatomical
regions also at epithelial cell resolution (Fig. 4c, d).

## Hassall's Bodies occupy a sizeable volume of the human thymic medulla

HBs are located in the medulla of the thymus, yet their role and time of
appearance remain debated. Literature on HBs appearance converges
towards GW15-19, displaying a high degree of variability. Thus, immu-
nofluorescence was used to identify HB structures that express keratin-10
(CK10) across human thymus development[8]. Representative images detect
earliest identification of few aggregates of CK10-positive cells at GW16 as
shown in Fig. 5a. From GW19, discs of fused CK10-positive cells are
observed, showing HB's structural organisation. In the later foetal stages, an
increase in HB number, volume, and structural complexity is observed. The
neonatal sample is particularly composed by large concentric circles,
characteristic of late cystic HBs[51].

To confirm that the structures observed in the medulla of PC-CT
slices were HBs, these were matched using downstream H&E of same

**Fig. 6 | Individually segmented Hassall's bodies.**
**a–d** Show four separately segmented HBs alongside their original position in SPC-CT slices of a foetal thymus (**e, f**); **g–j** show the same for the postnatal case, in which a single SPC-CT slice (**k**) was sufficient thanks to the increased HB presence. In both cases, a mild smoothing based on mesh interpolation was applied to avoid an excessively "blocky" appearance caused by the limited number of voxels an HB consists of.

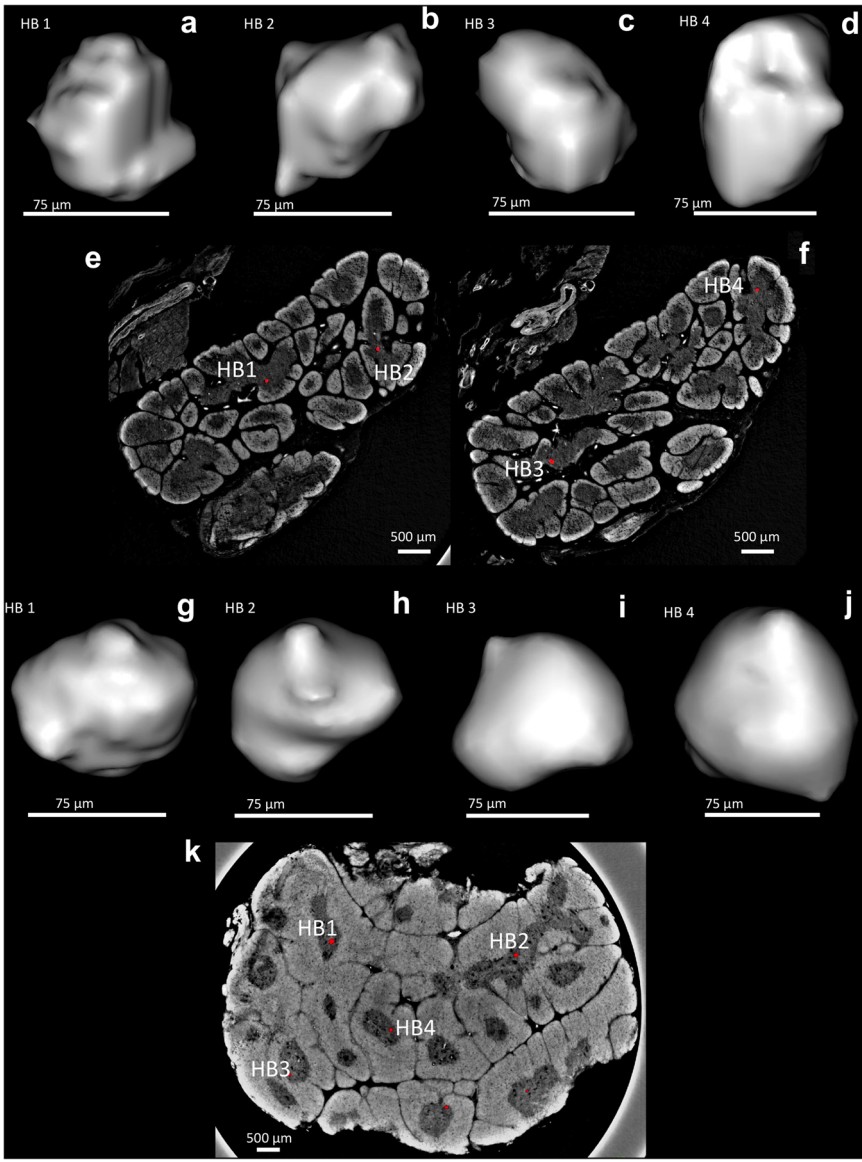

foetal and post-natal samples (Fig. 2). Histological matching of GW22 and 12MPN samples, once again showing dark-pink concentric structures in the medulla, confirmed that those structures observed in PC-CT slices were indeed HBs. These regions appear fully or partly filled with a radiologically denser material that might represent accumulation of epithelial reticular cells containing keratin filaments, some of which may go through dystrophic calcifications[8] (Fig. 5b). The cellular part of the HBs appears bright (possibly due to the high density of keratins), whilst the remainder is dark (as empty spaces are reported in histology). These were found in two foetal (GW19 and 22), and all postnatal thymi (Fig. 3; Supplementary Fig. S1).

Here, quantitative analysis of volumetric HB contents within the medullary compartment was achieved, as a function of developmental stage (Fig. 5c). Identification of HBs via PC-CT allowed to segment and render volumetrically these structures, for a better appreciation of their irregular shape (Fig. 6). Such irregularity explains the high degree of variability in results of quantification of HB areas in histological slices (Supplementary Fig. S5). HBs were identified in 8 out of 15 samples and progressively represented a sizeable percentage of the overall medulla volume, reaching approximately 25% after 4MPN. This suggests that HBs form an integral part of the thymic anatomy by constituting a third compartment, leaning towards the hypothesis of a vital role of these complex structures in the function of the organ.

## Access to SPC-CT comparable data using standard laboratory equipment

The possibility to assess whole organ development by performing volumetric and quantitative analysis allows gaining information that standard histology cannot provide. However, access to synchrotron facilities is limited and would prevent the application of PC-CT for the study of thymic changes in ageing or pathological conditions. Therefore, we set out to assess the capability of a lab-based system (namely EI) to correctly identify the basic components of the thymic anatomy as done with synchrotron-based PC-CT. To this aim we performed acquisition and comparison of the SPC-CT and H-CT slices of a 19-day old postnatal thymus (Fig. 7a, b, respectively). We note that, although the same sample was scanned, locating exactly the corresponding slice proved difficult, due to possible sample deformation occurring during transportation from the overseas synchrotron. This notwithstanding, the comparison reveals that the thymic components identified in the SPC-CT slices are also visualised in the corresponding H-CT slices, with resolution comparable to the synchrotron case. More specifically, we can distinguish between the cortex and medulla components as well as visualise HBs and BVs. Supplementary Fig. S6 shows that automatic segmentation approaches analogous to that used to extract quantitative volumetric measures from the SPC-CT datasets (see Supplementary Fig. S2) can also be applied to the H-CT data.

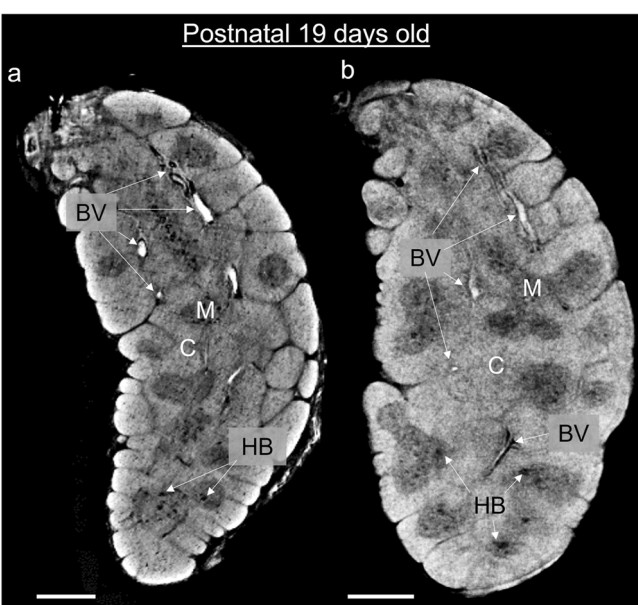

**Postnatal 19 days old**

**Fig. 7 | Laboratory-based CT solutions provide the same structural information as synchrotron equipment.** Corresponding slices of a 19-day post-natal thymus acquired using SPC-CT (**a**) and H-CT (**b**). Both depict the same structural components of this sample, including cortex (C), medulla (M), Hassall's bodies (HB) and blood vessels (BV). Scale bars: 500 μm.

## Discussion

Further to enabling the visualisation of internal anatomy of the thymus in arbitrary anatomical planes, the non-destructive nature of SPC-CT enabled us to generate what are, to the best of our knowledge, the first three-dimensional images of the human thymus, and to study the 3-dimensional evolution of the organ as a function of age. Specifically, we could: (1) appreciate the complex corticomedullary organisation of the entire human organ in 3D; and (2) provide the first volumetric visualisation of the HB morphology, with previous studies endeavouring to describe HBs having extracted morphological information only in a 2D fashion using essentially histology[18,27,28].

Our work demonstrates the high compatibility of SPC-CT slices with standard histology sections. This has allowed us to describe the 3D structure and quantify the volumes of different thymic anatomical compartments over human development and in postnatal stages that, until now, were described only on a single anatomical plane. Furthermore, the PC-CT protocol is compatible with downstream processing of the samples such as histological sections and staining and potentially spatial transcriptomics analysis.

The possibility to combine visualisation in 3D and quantification of volumes of the HBs has enabled us to define their progressive growth and quantify their occupancy of about 25% of the whole medulla in the paediatric thymus. Thus, HBs should be considered as a thymic compartment *per se*, in line with the increasing evidence of their role in immunity and cytokine production and in contrast with previous assumptions on their "non-functional" status[18,22,23].

In previous studies, thymic structural components were visually and quantitatively assessed across samples to infer biological conclusions and examine associated functional changes[25–28]. However, limitations in the assessment tools hindered the use of the extracted information to perform robust intra- and inter-sample comparisons. Examples of these limitations include: 1) extraction of information on a non-volumetric basis, with measures based on a single histological slice assumed to be representative of the entire volume; 2) use of non-standardised, subjective measurements for the quantification of structural components (e.g., manual measurements performed on histological slices); 3) lack of sample-specific normalisation for the extracted measures (e.g., inter-sample comparison performed assuming no variation in the physical size of the individual samples).

The application of the quantitative assessments to human thymi, coupled with volumetric sample visualisation based on PC-CT, allowed for a comprehensive assessment of thymic structural changes across the developmental and early postnatal periods. PC-CT was shown to extract information compatible to that provided by current gold standards such as histology and flow-cytometry, with the added advantage of doing this in a non-destructive three-dimensional fashion. PC-CT images are volumetric, normalised and based on global information over the entire scanned volume. An additional advantage associated with their use is the avoidance of problems caused by the distortion or loss of tissue that may occur during histological preparation, and of potential artifacts introduced during cutting, both of which can impact the interpretation of the tissue.

We observed that organ's division into lobules started at the foetal age of GW10[2], extended into the 11–22 GW age range and was clearly observed in 3D.

The difference in relative corticomedullary content between foetal and postnatal ages was confirmed quantitatively, based on the volumetric medulla content of all the samples considered in this work.

Finally, morphology of the HBs was visualised volumetrically. The volumetric content of these structures revealed an increase in the HBs content as a function of age.

Even though the presented information has been extracted from thymi over a limited age range, the results establish an initial benchmark for studying thymic changes not only during development and morphogenesis, but also in clinical conditions and during physiologic involution with age. The quantitative aspects underlying the 3D complexity of the thymus represents a reference for the structural changes occurring upon injury or disease. Volumetric information regarding Hassall's bodies becomes crucial in thymic pathologies such as Down syndrome where immunohistochemical staining showed altered architecture of the thymic medulla with increased frequencies of AIRE- positive medullary epithelial cells as well as enlarged HBs[52,53]. In addition, volumetric analysis will be helpful to explain the heterogeneity of thymic epithelial tumours where the organ architecture is compromised. While so far this has been studied only on 2D histological sections looking at few markers, 3D volumetric analysis might reveal new insights regarding the structural changes happening and help to improve classification of the highly heterogenous thymic neoplasms[54,55].

Furthermore, this technology will aid the current tissue engineering approaches that aim at reconstructing functional and structural complexity of functional organs such as thymus.

Going forward, lab-based PC-CT could play a pivotal role in guiding the protocol's progression through more advanced testing stages providing with new diagnostic tools.

## Data availability
All source data underlying the figures is included in the Supplementary Data. Additional data pertaining to the acquired images can be made available upon reasonable requests to the corresponding author.

## Code availability
The ML-based segmentation software is described in ref. 47 and available from the link https://github.com/aleatzeni/SmartInterpol. The code for Paganin-based phase retrieval applied to synchrotron images was made available by the ESRF (see ref. 39). The code for phase retrieval of the laboratory images was developed in-house and is specific to the imaging system prototype developed at UCL; as such, it would require significant adaptation to be used on images produced by a different imaging system. This notwithstanding, it can be made available upon reasonable requests to the corresponding author. Image manipulation, rendering and 3D reconstruction were performed using open access software as per the cited references (ASTRA toolbox, refs. 40–42, Fiji, ref. 43, and Drishti, refs. 44–46).

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

## Acknowledgements

A.O. is supported by the Engineering and Physical Sciences Research Council (EPSRC - EP/T005408/1) and by the Royal Academy of Engineering under the Chairs in Emerging Technologies scheme (CiET1819/2/78). S.S. is a UKRI-EPSRC Doctoral Prize Fellow (EP/T517793/1). P.B. is supported by the European Research Council (ERC-StG Agreement No. 639429), the Rosetrees Trust (M362-F1; M553), the London Advanced Therapies – Research England (C2N-AT.006), the MRC Confidence in Concept scheme (Award: MC_PC_17180), the Innovate UK (Smart grant n.10005465), the NIHR GOSH Biomedical Research Centre and the Francis Crick Institute which receives its core funding from Cancer Research UK (CC0102), the UK Medical Research Council (CC0102), and the Wellcome Trust (CC0102). R.R. is supported by a Marie Skłodowska-Curie Individual Fellowships (MSC-IF No. 896014). V.C.R. is supported by a London Interdisciplinary Biosciences Consortium (BBSRC) fellowship. We thank Mark Turmaine for processing fixed samples and Marco Catucci for helping in collecting and processing the thymic tissues; Sahira Khalaf and Pei-Shi Chia for consenting patients for tissue donation. We thank Dr. Bianca De Blasi for the fruitful discussions. We are grateful to Kirsten Nowlan for critical reading of the manuscript.

## Author contributions

A.O. and P.B. designed the study. S.S performed the data analysis for generating the PC-CT results. S.S and L.M performed the scans with the lab-based EI PC-CT system. O.K.O generated the histological slices. R.R performed the flow cytometry data acquisition and analysis. R.R. and V.C.R. performed immunofluorescence and confocal image analyses. A.A. developed the machine learning tool used for this project and assisted for its optimisation and analysis of the related PC-CT datasets. TP performed the segmentation of the lab-based H-CT dataset. S.C. was involved in collection, preparation and processing of the thymic tissues used for SPC-CT imaging and histological analysis. P.B. processed the thymic tissues for the SPC-CT analysis. N.J.S. and P.D.C. contributed to data analysis and result interpretation. S.S, A.O, P.B, J.C.H, F.A.V, A.B and A.M conceived and performed the experiments at the ESRF. S.S., R.R., V.C.R., P.B and A.O wrote the manuscript.

## Competing interests

The authors declare no competing interests.

## Additional information

[1]Department of Medical Physics and Biomedical Engineering, University College London, London WC1E 6BT, UK. [2]Epithelial Stem Cell Biology & Regenerative Medicine laboratory, The Francis Crick Institute, London NW1 1AT, UK. [3]Institute of Immunity & Transplantation, Division of Infection & Immunity, UCL, London NW3 2PP, UK. [4]Department of Histopathology, Great Ormond Street Hospital for Children NHS Foundation Trust, London WC1N 1EH, UK. [5]Centre for Medical Image Computing, Department of Medical Physics and Biomedical Engineering, University College London, London WC1E 6BT, UK. [6]Stem Cell and Regenerative Medicine Section, Great Ormond Street Institute of Child Health, University College London, London WC1N 1EH, UK. [7]Specialist Neonatal and Paediatric Surgery, Great Ormond Street Hospital NHS Trust, London, UK. [8]European Synchrotron Radiation Facility, Grenoble 38043, France. [9]Dept. of Physics "G. Occhialini", University Milano Bicocca, Milano, Italy. [10]Present address: ENEA - Radiation Protection Institute, Via Martiri di Monte Sole 4, 40129 Bologna, Italy. [11]Present address: The Roger Williams Institute of Hepatology, 111 Coldharbour Lane, SE5 9NT London, UK. [12]Present address: Advanced Photon Source, Argonne National Labs, Lemont, IL, USA. [13]These authors jointly supervised this work: Paola Bonfanti, Alessandro Olivo. ✉e-mail: paola.bonfanti@crick.ac.uk; a.olivo@ucl.ac.uk

