## [Peer Review File · Communications Medicine]

Reviewers' comments:

Reviewer #1 (Remarks to the Author):

This manuscript describes that the phase-contrast x-ray micro-CT allows whole-organ volumetric and architectural analysis of corticomedullary structure and Hassall's corpuscles in the human thymus. Whole-organ imaging analysis of the human thymus is important towards better understanding of various immunological disorders. However, what is new and 'advanced' in the technology described in this manuscript reads unclear. In addition, this manuscript only verifies previously reported findings on embryonic development of the human thymus and offers no new implications in human physiology or pathology. This manuscript contains further problems as follows.

Page 2-4: The authors suggest that volumetric qualitative and quantitative analysis is lacking in the field of thymus research. However, there are many studies on the computed tomography imaging of the thymus from various species including the human. The authors may want to cite those papers (e.g., Bogot and Quint, *Cancer Imaging* 2005; Sakata, et al., *J Immunol* 2018; Liu, et al., *Sci Rep* 2022; Cordella, et al. *Vet Sci* 2023) and clarify what is new here.

Fig 1-5, page 5 and so on: How the authors identified the cortex, medulla, and Hassall's corpuscles is not specified. Molecular markers for cTECs v mTECs (e.g. Krt8, Krt5, Psmb11, Aire) and immature v mature thymocytes (e.g. CD4/CD8, TCR) may help.

Fig. 3: The lack of CD205 detection in fetal thymus may only reflect a lower expression or higher enzyme-susceptibility of CD205 in embryonic cTECs than postnatal cTECs (Shakib, et al. *J Immunol* 2009). The cortex/medulla contents should be more carefully evaluated.

Reviewer #2 (Remarks to the Author):

The manuscript titled "Investigating Thymic Microarchitecture Using Phase-Contrast X-ray Micro-CT" attempts to elucidate the intricate structure and function of the human thymus. The manuscript is well-written and presents an innovative approach to studying thymic structure. The authors introduce phase-contrast x-ray micro-CT to analyze thymic architecture across developmental stages. The study's methodology is sound, and the presentation is clear, highlighting the potential of phase-contrast x-ray micro-CT for such investigations. Notably, the accessibility of this innovative approach, including the possible utilization of a more compact setup (like the one developed and presented by the authors), has promising implications for widening its availability within the research, medical, and biomedical communities. Overall, the manuscript makes a valuable contribution by examining thymic microarchitecture.

However, there are some critical points that need to be addressed.

1) The choice of 300 slices for segmentation and subsequent analysis raises questions. Why this specific subset instead of the entire volume, particularly when the estimation is volume-normalized? Is the selected volume truly representative? How was this subset determined apart from matching the same number of slices of the smaller sample? Is the distribution of cortex and medulla of isotropic type? These slices were selected always in the same region for all samples? This information is important because it can be a potential source of bias. Given the segmentation's reliance on machine learning, considering applying it to the complete sample could lead to a more precise estimate and impartial comparison, at least for the medulla volumetric percentage content. Alternatively, demonstrating the compatibility of the two estimates (full volume and 300 slices) within error margins, at least for a small range of diverse age cases, would be essential.

2) Regarding the manual segmentation of HBs, could you clarify the individuals responsible for the segmentation process? Was it performed by an expert pathologist under his/her guidance? Could the authors consider including, as supplementary content, diverse examples of HB segmentation? Utilizing the images already provided (for example showing the overlay of a portion of the CT slice and corresponding segmented structures) in the study could provide additional insight. It's noteworthy that in histological representation, as well as in existing literature, HBs exhibit a somewhat irregular shape when observed in a single plane. Does the spherical assumption reported in the manuscript refer to very bright elements inside HBs or does it refer to their 3D appearance? Within the comparison between CT scans and histology, authors reported bright regions sometimes surrounded by dark regions. When manually segmenting HBs, what was included in the selection? From the visual comparison of panels in Figure 4d, the dark region seems to belong to the HB and not rather be an empty space. Was it included in the manual segmentation? Are the gray values of that region compatible with empty spaces (air) or does the visual appearance just depend on the contrast that was set?

3) The presence of intensely bright spots might indicate the presence of calcifications. Previous studies have reported that HBs can lead to calcifications under acute stress conditions. To gain more clarity, would it be feasible to collaborate with pathologists and explore additional complementary techniques? It could help confirm whether these spots are indeed calcifications or, as the authors hypothesize, a region characterized by a high density of keratin. Additionally, in Figure 4c, the histological section reveals an absence of tissue corresponding to the brightly illuminated area in the CT scan. This observation inclines me towards considering calcification as the possible explanation, given that calcifications tend to dislodge during histological processing (sectioning of the tissue with microtome).

4) Providing a clearer description of the flow cytometry data would help readers, especially those unfamiliar with the technique, understand the significance of the results. While the comparison does show noticeable differences between foetal and post-partum sample data trends, adding more details could make the interpretation even more accessible.

5) I find the method for determining the volume percentage content of the medulla and HBs unclear based on equations 1.1 and 1.2. My expectation was for a volume fraction, calculated as the ratio of the specific quantity's volume (in voxels or mm^3) to the total enclosing volume (also in voxels or mm^3). However, it appears that individual voxel grayscale values are somehow weighted, which contrasts with the expected approach. There might be an issue with the formula provided, prompting my request for either clarification or correction if necessary. I must admit that I'm rather puzzled by the definition of: "Sum Pixel Value". If the consideration indeed involves grayscale values rather than voxel sizes, I'd appreciate insight into the reasoning behind this choice. Additionally, could the identified growth trend be influenced by this approach (due to the potential increase in high-intensity bodies)? Assuming grayscale values are being weighted, could the authors explore whether estimating the medulla and HBs fraction based on volume ratios yields the same trend? I suspect there could be a consistent misdefinition concerning the metrics employed. The notation 'Sum Pixel Value' is utilized across all three equations, which seems somewhat ill-suited, particularly when estimating the FACS-based medulla content, given its immediate definition before equation 1.3.

6) Are the two sections reported in Figure 5 depicting the same sample for both acquisition modes? If so, could you provide insight into the reasoning for not using the same section in both cases? This decision seems to impact the direct comparison between the techniques. While I understand the intention of showcasing the ability of H-CT to capture information comparable to a synchrotron light imaging system, it might be more effective to directly compare identical sections. Could you elaborate on the rationale behind this choice or, alternatively, consider presenting images of the same section? Such an approach could enhance the clarity and impact of the comparison.

7) Reference No. 28 appears incongruous, considering the discrepancy in imaging setups. I propose either its removal or replacement with a suitable alternative. Assuming the imaging system aligns with the description in reference 29 (PCO Edge with a native pixel size of 6.5 μm), a 2x magnification would result in a pixel size of 3.25 μm or less, assuming a magnification strictly greater than 1. While this detail might mainly affect the scale bars of the CT images, if they are inaccurate, I recommend correcting them. Additionally, could you provide the propagation distance between the sample and the detector? Concerning phase retrieval, was a quantitative approach employed, specifying the delta/beta ratio for a particular material pair (if so, which materials and delta/beta ratio), or was it adaptively adjusted to attain the desired image quality?

8) In the following sentence of the Introduction:

“As currently reported, Hassall’s bodies (HBs) (or Hassall’s corpuscles) are formed by around GW 154 these are unique to the medulla⁵ and are commonly described as “concentric bodies” composed of epithelial reticular cells filled with keratin filaments⁶”

There is, probably, a full stop missing.

9) To confirm the HB nature of certain structures shown in Figure 3b, it is recommended to use a histochemical marker (pancytokeratin for example) that would highlight only corpuscles. With H&E, it is difficult to distinguish a corpuscle from a blood vessel. Even if 3D can exclude the vascular nature by following the structure in successive planes, the slice-based comparison would need further proof of the nature of the structure identified.

Reviewer #3 (Remarks to the Author):

All in all the paper is clear to read, and shows a nice application of X-ray imaging towards 3D characterization of tissues instead of just 2D histological slices. This is a topic that has been under progress for several decades now, but still these techniques are not in routine use in the clinics. By showing that lab based equipment can be used for such studies, and also using semi-automated analysis techniques, the paper presents a step towards easier adoption of such techniques in practice.

Here are some minor comments about the techniques used and the presentation of the results:

page 4, end of second paragraph: there are many different pc techniques, with widely varying resolutions. Here you talk like 3.5 μm voxel size is the feature of the technique, rather than the specific setup that was used.

figure 5: The slices look very different, why not show the exactly same slice for both? Has the sample deformed in the meantime or what is this?

page 16, Pre-imaging sample preparation: why not to use hydrated samples for imaging? After all, this is one of the main selling points of phase contrast.

page 18: for sample large than the detector's field of view, did you scan in multiple parts to cover the whole sample? If not, how did you choose the ROI?

page 20, first line: how did you train the ML model? did you label some slices (for one sample, or for each sample), or did you use readily trained model as such? It is not clear how to reproduce this step.

page 20, equations 1.1 - 1.3: to me it seems that eq 1.1, 1.2 and 1.3 are not strictly necessary, as these calculations are rather trivial, and they are already well explained in the text.

page 20, HB segmentation: as the HBs are rather small in size, there is chance that choosing the segmentation somewhat differently would lead to quite different results. it would be good to indicate how reliable the HB volumes are.

signed,

Heikki Suhonen

We would like to thank the Reviewers for their useful insights, the implementation of which has led to a much better manuscript. In the following, we provide a point-by-point response to all their comments and list all changes to the manuscript made as a result. We hope the arguments brought in response to Reviewer 1, and especially the changes made to the manuscript as a result, will also address the Editor's comments re. advance vs. existing literature and potential clinical implications of our work, but we remain of course at the Editor's disposal should they deem that more needs to be done on those aspects. For ease of visualization, new text has been highlighted in red in the revised manuscript.

Reviewer 1

1. This manuscript describes that the phase-contrast x-ray micro-CT allows whole-organ volumetric and architectural analysis of corticomedullary structure and Hassall's corpuscles in the human thymus. Whole-organ imaging analysis of the human thymus is important towards better understanding of various immunological disorders. However, what is new and 'advanced' in the technology described in this manuscript reads unclear. In addition, this manuscript only verifies previously reported findings on embryonic development of the human thymus and offers no new implications in human physiology or pathology. This manuscript contains further problems as follows.

We thank the Reviewer for stating that “Whole-organ imaging analysis of the human thymus is important towards better understanding of various immunological disorders”. We appreciate that in the first version of the manuscript the novelty of the technology developed was not sufficiently highlighted. We have now better explained both the technical and biological advancements and discussed their implication in the field and potential impact in medical applications. The manuscript has now been heavily modified in the text and re-organized to improve readability. All changes are marked in red.

We consider our findings of high potential interest as they describe thymic anatomical compartments through fetal development till postnatal stages with relative volumetric quantification and 3D visualization; these data are unprecedented as no Phase Contrast Computed Tomography (PC-CT) of thymi was reported before -neither synchrotron nor lab-based- as this is *not* conventional CT. We discuss this further in point 2 below. Furthermore, our data are acquired on human samples starting as early as Carnegie Stage (CS) 23 through 1.5-year-old, far beyond the developmental stages.

Implications for human physiology and pathology are several. Whole organ volumetric acquisition may allow to overcome limitations in the assessment tools hindered by the extraction of information by measurements on thin histological sections considered ‘representative’ of the whole volume. For instance, structural changes during embryonic thymus development in inborn errors of stromal defects might be better evaluated in those cases where only whole organ size is the feature reported so far, and no precise cortical/medullary quantification can currently be carried out. There is an increasing amount of diagnosis of thymic defects which remain to be fully characterized (Kreins et al., 2021).

We report also the first volumetric quantification of the progressive growth of HBs' volume that reaches about 25% of the entire medulla in postnatal thymus. This infers a major role of HBs in thymus function that has been neglected till very recently and we now propose it as a *bona fide* functional compartment of the human thymus whose detailed composition, complexity and functions must be further investigated. Indeed, ‘enlarged’ HBs have been suggested to be a feature in diseased thymi such as Down Syndrome with autoimmune phenotype (Skogberg G et al., 2014; Marcovecchio GE et al., 2021). Cystic enlargement of HBs was reported also on the basis of acquired inflammatory changes of the thymus (Suter et al., 1991). Another potential application is the possibility to classify the highly heterogeneous thymic tumors which present alteration of thymic architecture and rely on a complex histological classification; finally, this technology will help defining the margins of resection for those tumors infiltrating the surrounding tissues, to reduce the risk of missing some crucial areas

via rough selection based on the macroscopic appearance. Several of these examples have now been added in the Discussion section of the revised manuscript (please see parts in red).

2. Page 2-4: The authors suggest that volumetric qualitative and quantitative analysis is lacking in the field of thymus research. However, there are many studies on the computed tomography imaging of the thymus from various species including the human. The authors may want to cite those papers (e.g., Bogot and Quint, *Cancer Imaging* 2005; Sakata, et al., *J Immunol* 2018; Liu, et al., *Sci Rep* 2022; Cordella, et al. *Vet Sci* 2023) and clarify what is new here.

SPC-CT and lab-based PC-CT technologies are very different from standard diagnostic CT scans. While the latter is purely based on x-ray attenuation, the former use the phase changes that x-rays undergo when traversing different tissues to generate image contrast, which is a *physically different* mechanism. An effective way to express x-rays interaction with tissue is by means of the complex refractive index $n = 1 - \delta + i\beta$, where δ and β are largely independent quantities. Standard diagnostic CT only uses β , while SPC-CT and lab-based PC-CT use δ . δ is much more sensitive to the faint changes that take place in soft tissues, so its use leads to unprecedented soft tissue sensitivity for x-rays, revealing features and details that are completely undetectable to conventional x-rays. Although not clinically available yet, phase contrast x-ray methods are established research tools, which is why we provide the specific implementations of the phase contrast techniques we have used in the methods section, rather than explaining phase contrast from scratch - for which the reader is referred to the cited references. However, we would be happy to add more details on the basics of phase contrast x-rays if the Reviewer so wishes.

Indeed, the references cited by the Reviewer report data unrelated to our work and methodology:

- Bogot and Quint, *Cancer Imaging* 2005 and Liu, et al., *Sci Rep* 2022: these papers report standard diagnostic CT scan in live patients, not related to the type of resolution and technology addressed in our work.
- Sakata, et al., *J Immunol* 2018: refers to mouse histology, not related to our approach.
- Cordella, et al. *Vet Sci* 2023: refers to dog CT scans, not related to our approach.

3. Fig 1-5, page 5 and so on: How the authors identified the cortex, medulla, and Hassall's corpuscles is not specified. Molecular markers for cTECs v mTECs (e.g. *Krt8*, *Krt5*, *Psmb11*, *Aire*) and immature v mature thymocytes (e.g. *CD4/CD8*, *TCR*) may help.

We have identified the compartment of the thymus by histological evaluation by H&E. The compartments of the thymus (i.e., cortex, medulla and HBs) are routinely identified (both in the clinic and research labs) by their histological features also without molecular marker detection. H&E histology defines the cortex stain as more basophilic (nucleic acids are stained purplish by hematoxylin) than medulla because it contains a higher number of closely packed lymphocytes. HBs are unique structures to the thymus medulla and are characterized by epithelial cells arranged in concentric layers that have keratinized.

Reference examples on how the thymic compartments are described histologically and correlate to molecular markers are many in the literature; some examples are represented by Hale, 2004 (<https://doi.org/10.1016/j.anndiagpath.2003.11.006>) and Hale et al., 2020 (<https://journals.plos.org/plosone/article/figures?id=10.1371/journal.pone.0230668>), demonstrating correspondence between these methodologies that however aim at addressing different types of questions. In fact, molecular markers define specific sub-populations of each of these compartments and they are of no use for whole quantification of thymic compartment size and macroscopic changes. In fact, most of cellularity in the thymus is represented by developing thymocytes while stromal/epithelial cells are less than 2% (Park et al., 2020; Ragazzini et al., 2023). Nevertheless, we have now added molecular detection for identification of HBs (see below, point 9 to Reviewer 2) and of cortex and medulla (see next point 4).

4. Fig. 3: The lack of CD205 detection in fetal thymus may only reflect a lower expression or higher

enzyme-susceptibility of CD205 in embryonic cTECs than postnatal cTECs (Shakib, et al. *J Immunol* 2009). The cortex/medulla contents should be more carefully evaluated.

We agree with the Reviewer that CD205 immunostaining during early phase of development may have a lower level of expression on cTEC and therefore the FACS quantification may underestimate the number of cTEC. We have stained the compartment of cortex in early stages of fetal development by IHC against CD205 and provide below the **Figure 1 for the Reviewer** confirming that CD205 is already expressed at early developmental stages. These images confirm correlation of CD205 cTEC staining at IHC and flow cytometry. Flow cytometry allows to discriminate the expression of CD205 between epithelial (CD45 negative) and dendritic cells (CD45 positive) and therefore to correctly quantify the epithelial cells in each compartment.

Figure 1: Representative immunofluorescence images depicting cTEC (CD205, green) and mTEC (KRT14, red) across development. Nuclei were counterstained with Hoescht. Red blood cells autofluorescence, as observed in the perivascular space. CD205 positive cells and KRT negative in the medulla (white arrows) represent CD205 positive dendritic cells. KRT14 stains also the subcapsular layer in the cortex, thus supporting that molecular markers are not necessarily compartment-specific. Scale bar: 50µm.

Finally, we have now added H&E comparison in Fig.1b and quantification of relative medullary area across development in Fig.4b.

Reviewer 2

1. The choice of 300 slices for segmentation and subsequent analysis raises questions. Why this specific subset instead of the entire volume, particularly when the estimation is volume-normalized? Is the selected volume truly representative? How was this subset determined apart from matching the same number of slices of the smaller sample? Is the distribution of cortex and medulla of isotropic type? These slices were selected always in the same region for all samples? This information is important because it can be a potential source of bias. Given the segmentation's reliance on machine learning, considering applying it to the complete sample could lead to a more precise estimate and impartial comparison, at least for the medulla volumetric percentage content. Alternatively, demonstrating the compatibility of the two estimates (full volume and 300 slices) within error margins, at least for a small range of diverse age cases, would be essential.

The Reviewer makes a valid point, and indeed one we hadn't initially considered. The choice of 300 slices was dictated by the size of the smallest available samples; we then decided to stick to the same number of slices for all samples for consistency (this is now explicitly explained in the methods

section: “This was the maximum number of slices available in the smallest samples, and we opted to use the same number of slices for all samples for consistency, by selecting the central 300 slices in larger samples”).

To take this important point properly into account, we have performed additional analysis along the lines suggested by the reviewer. We have selected a foetal and a postnatal thymus, repeated the analysis on their entire volumes (900 and 1200 slices respectively), and compared the results with those extracted from the central 300 slices. As can be seen from the results (reported in Suppl. Fig. 3 and Suppl. Table 2), virtually identical outcomes are obtained – the reviewer is referred to the revised suppl. mat. section for details. The following text was added to the main manuscript to address this point:

“To show that our choice of using the central 300 slices does not affect the final results, we have repeated the above analysis on the entire volume for a foetal (900 slices) and a postnatal (1200 slices) thymus, and compared the extracted results to those obtained from the central 300 slices. The results, presented in supplementary figure 3 and supplementary table 2, demonstrate that practically identical outcomes are obtained in the two cases.”

2. Regarding the manual segmentation of HBs, could you clarify the individuals responsible for the segmentation process? Was it performed by an expert pathologist under his/her guidance? Could the authors consider including, as supplementary content, diverse examples of HB segmentation? Utilizing the images already provided (for example showing the overlay of a portion of the CT slice and corresponding segmented structures) in the study could provide additional insight. It's noteworthy that in histological representation, as well as in existing literature, HBs exhibit a somewhat irregular shape when observed in a single plane. Does the spherical assumption reported in the manuscript refer to very bright elements inside HBs or does it refer to their 3D appearance? Within the comparison between CT scans and histology, authors reported bright regions sometimes surrounded by dark regions. When manually segmenting HBs, what was included in the selection? From the visual comparison of panels in Figure 4d, the dark region seems to belong to the HB and not rather be an empty space. Was it included in the manual segmentation? Are the gray values of that region compatible with empty spaces (air) or does the visual appearance just depend on the contrast that was set?

The manual segmentation of the HBs was performed by SS under the supervision of an experienced biomedical researcher with specific expertise on the thymus (PB, co-supervisor of the entire study with AO). Segmentation was based on morphology, and included both the dark and the bright (where present) regions as hinted by the reviewer; this is now clearly stated in the methods section (“**Manual segmentation was performed by SS under PB’s supervision; it was based on morphology, and both dark and bright (where present) regions were included.**”). We did not intend to make a “spherical assumption” for the HB shape, although in hindsight we understand why the text as written could have given this impression; with “spherical” we meant to indicate a self-contained, localized volume, as opposed to a longitudinally extended one that could be confused with e.g., a blood vessel. To address this, we have removed the reference to “spherical” from the caption of former Fig. 2 (now Fig. 3), and replaced “spherical” with “**irregular**” in the main text. Most importantly, we have introduced a new figure (Fig 6) where individually segmented HBs are shown, alongside their original position in one of the SPC-CT slices that intersects them. Reference to this figure is made in the caption of Fig. 3 (previously Fig 2) (“**a more detailed visualisation of their shape is provided in Figure 6**”) and in the main text (“**Identification of HBs via PC-CT allowed to segment and render volumetrically these structures, for a better appreciation of their irregular shape (Fig. 6)**”). We hope this addresses the main action requested by the Reviewer, but remain of course at their disposal should they consider that we have missed anything.

3. The presence of intensely bright spots might indicate the presence of calcifications. Previous studies have reported that HBs can lead to calcifications under acute stress conditions. To gain more clarity, would it be feasible to collaborate with pathologists and explore additional complementary techniques? It could help confirm whether these spots are indeed calcifications or, as the authors

hypothesize, a region characterized by a high density of keratin. Additionally, in Figure 4c, the histological section reveals an absence of tissue corresponding to the brightly illuminated area in the CT scan. This observation inclines me towards considering calcification as the possible explanation, given that calcifications tend to dislodge during histological processing (sectioning of the tissue with microtome).

We thank the Reviewer for highlighting the potential presence of calcifications in the HBs and we agree this is a possible interpretation of some of the bright spots and potential loss/dislodge during sectioning for H&E. Given that calcifications are not a constant feature of all HBs (Mills SE, Histology for pathologists, 4th edition) and given that calcification staining would require a very different preparation of samples, it is not possible to conduct calcification-specific staining on sections of the same samples used for SPC-CT.

Therefore, we have adapted the text and added this possibility that is discussed in the main text as reported here (“**These regions appear fully or partly filled with a radiologically denser material that might represent accumulation of epithelial reticular cells containing keratin filaments, some of which may go through dystrophic calcifications**”). We also take the opportunity to highlight how the heterogeneity and complexity of HBs is an under investigated aspect of human thymus, that has been conducted in old literature and reported mainly in textbooks solely based on histological analysis. We hope that our work which highlights these aspects in addition to the volumetric resolution of this compartment will stimulate further research on their structures, molecular features and immunological functions.

4. Providing a clearer description of the flow cytometry data would help readers, especially those unfamiliar with the technique, understand the significance of the results. While the comparison does show noticeable differences between foetal and post-partum sample data trends, adding more details could make the interpretation even more accessible.

We agree with the Reviewer that the description of flow cytometry data in the previous version of the manuscript was inadequate. We have now changed the text in the revised manuscript. Volumetric measurements of cortical and medullary compartments provide an accurate way to compare samples across the developmental trajectory considering whole cellular sub-populations. See also point 4 for Reviewer 1. To understand whether epithelial cell numbers adapt to cortex and medulla volumetric changes during development, we performed flow cytometry analysis on dissociated thymic samples. We focused on stromal population (CD45-neg) and analysed expression of cortical (EPCAM^{low}CD205^{pos}) and medullary (EpCAM^{pos}CD205^{low}) epithelial populations to determine their abundance. We observed that also at epithelial cell resolution foetal thymi (n=9) have significantly higher medulla content (78 ± 1) than the postnatal ones. Therefore, the volumetric changes do not reflect only an increase/decrease of thymocytes, and correlate with structural changes of both compartments.

5. I find the method for determining the volume percentage content of the medulla and HBs unclear based on equations 1.1 and 1.2. My expectation was for a volume fraction, calculated as the ratio of the specific quantity's volume (in voxels or mm³) to the total enclosing volume (also in voxels or mm³). However, it appears that individual voxel grayscale values are somehow weighted, which contrasts with the expected approach. There might be an issue with the formula provided, prompting my request for either clarification or correction if necessary. I must admit that I'm rather puzzled by the definition of: “Sum Pixel Value”. If the consideration indeed involves grayscale values rather than voxel sizes, I'd appreciate insight into the reasoning behind this choice. Additionally, could the identified growth trend be influenced by this approach (due to the potential increase in high-intensity bodies)? Assuming grayscale values are being weighted, could the authors explore whether estimating the medulla and HBs fraction based on volume ratios yields the same trend? I suspect there could be a consistent misdefinition concerning the metrics employed. The notation 'Sum Pixel Value' is utilized across all three equations, which seems somewhat ill-suited, particularly when estimating the FACS-based medulla content, given its immediate definition before equation 1.3.

We understand the Reviewer's concern, which however arises from a misunderstanding due to a bad choice of words from our side – in particular, we should not have used the expression “*Sum Pixel Value*”. All volume fraction calculations are performed on the *binary mask images*, i.e. pixel (actually *voxel*) content is either one or zero: hence this corresponds exactly to the “*volume fraction, calculated as the ratio of the specific quantity's volume (in voxels or mm³) to the total enclosing volume (also in voxels or mm³)*” suggested by the reviewer, and no weighting whatsoever is assigned to the pixel grey values. Since the text already makes direct reference to the binary masks (“**The extracted cortex and medulla binary masks were used for obtaining the volumetric medulla content for each sample**”), our impression is that this point should be addressed simply by switching “*Sum Pixel Values*” into “*Sum voxels*” in all equations, but we remain at the Reviewer's disposal should they see some remaining ambiguity in other parts of the text. This should also address the Reviewer's following comments about the possible impact of using grey scale values instead of just volumes, since it clarifies that no grey values were used; we apologize for having used an unclear expression in the original submission.

6. *Are the two sections reported in Figure 5 depicting the same sample for both acquisition modes? If so, could you provide insight into the reasoning for not using the same section in both cases? This decision seems to impact the direct comparison between the techniques. While I understand the intention of showcasing the ability of H-CT to capture information comparable to a synchrotron light imaging system, it might be more effective to directly compare identical sections. Could you elaborate on the rationale behind this choice or, alternatively, consider presenting images of the same section? Such an approach could enhance the clarity and impact of the comparison.*

Images shown in Fig 5 (NB now Fig 7) are taken from the same *sample*, but do not represent exactly the same *slice*. The samples had to be transported from our labs in London to the ESRF in France and back, which inevitably resulted in some degree of compression, deformation, etc: locating exactly the same slice proved practically impossible, possibly because what was a flat slice in one scan became a “bent” one in the second. We would like to note that our goal was not to demonstrate exact correspondence between synchrotron and lab scans (the former will inevitably retain a better quality), but rather that the same basic constituents of the thymic anatomy could also be identified in a lab scan. To address this comment, we removed the original reference to “assessing compatibility” between synchrotron and lab, and replaced it with “**assess the capability of a lab-based system (namely EI) to correctly identify the basic components of the thymic anatomy as done with synchrotron-based PC-CT**”; moreover, we added the sentence “**We note that, although the same sample was scanned, locating exactly the same slice proved impossible, due to possible sample deformation occurring during transportation from the overseas synchrotron**”.

7) *Reference No. 28 appears incongruous, considering the discrepancy in imaging setups. I propose either its removal or replacement with a suitable alternative. Assuming the imaging system aligns with the description in reference 29 (PCO Edge with a native pixel size of 6.5 μm), a 2x magnification would result in a pixel size of 3.25 μm or less, assuming a magnification strictly greater than 1. While this detail might mainly affect the scale bars of the CT images, if they are inaccurate, I recommend correcting them. Additionally, could you provide the propagation distance between the sample and the detector? Concerning phase retrieval, was a quantitative approach employed, specifying the delta/beta ratio for a particular material pair (if so, which materials and delta/beta ratio), or was it adaptively adjusted to attain the desired image quality?*

We agree with the Reviewer and are grateful to them for pointing this out, which was an oversight from our side. To address this comment, we have removed reference 28 which we agree is out of context, and replaced original reference 29 with one that reflects the lens system actually used in this case. We confirm that the effective pixel size was 3.5 micron, and have amended the corresponding sentence as follows: “**via a 1.75x lens system³⁹ yielding an effective pixel size of 3.7 μm, which is slightly demagnified at the sample by the moderate beam divergence introduced by the bent Laue monochromator; the effective pixel size at the sample was measured experimentally and found to be of approximately 3.5 μm**”. Please note that the significantly

different reference number (39 instead of 29) is due to the large number of new references added in response to various comments by the reviewers. The propagation distance was added (“**the sample-to-detector distance was of approximately 3.45 m**”, apologies for the omission) and, regarding Paganin retrieval, the Reviewer is correct that this was iteratively adapted to maximize image quality; the following sentence was therefore added: “**by adaptively adjusting the Paganin parameter until the desired level of image quality was obtained.**”.

8) *In the following sentence of the Introduction:*

“As currently reported, Hassall’s bodies (HBs) (or Hassall’s corpuscles) are formed by around GW 154 these are unique to the medulla⁵ and are commonly described as “concentric bodies” composed of epithelial reticular cells filled with keratin filaments⁶”

There is, probably, a full stop missing.

Indeed, thanks for spotting this, a full stop has now been introduced (“...around GW 15⁴. These are unique...”).

9) *To confirm the HB nature of certain structures shown in Figure 3b, it is recommended to use a histochemical marker (pancytokeratin for example) that would highlight only corpuscles. With H&E, it is difficult to distinguish a corpuscle from a blood vessel. Even if 3D can exclude the vascular nature by following the structure in successive planes, the slice-based comparison would need further proof of the nature of the structure identified.*

We have now added the immunohistochemistry for KRT10, a marker expressed exclusively by HBs (please see panel a in new Figure 5).

Reviewer 3

1. *page 4, end of second paragraph: there are many different pc techniques, with widely varying resolutions. Here you talk like 3.5 μm voxel size is the feature of the technique, rather than the specific setup that was used.*

The Reviewer is of course right. We have changed this sentence into “**PC-CT generates high-resolution images, the specific value of which depends on the used setup (e.g. 3.5 μm voxel size in the synchrotron-based part of this study)**”.

2. *figure 5: The slices look very different, why not show the exactly same slice for both? Has the sample deformed in the meantime or what is this?*

Please see response to point 6 by Reviewer 2, who raises exactly the same comment.

3. *page 16, Pre-imaging sample preparation: why not to use hydrated samples for imaging? After all, this is one of the main selling points of phase contrast.*

The Reviewer is correct that in principle we could have imaged hydrated samples; however, we chose the safest possible sample preparation method in view of the transportation from London to the ESRF in France and back. We were also hoping this would have ensured more consistency across volumes when samples were re-scanned with different systems, but unfortunately this turned out not to be true, as discussed in response to the previous point. To address at least in part this comment, we have added the following sentence: “**While in principle imaging hydrated samples is possible, we felt this was a safer preparation protocol in view of the need to transport the specimens to an overseas synchrotron and back**”.

4. *page 18: for sample large than the detector's field of view, did you scan in multiple parts to cover the whole sample? If not, how did you choose the ROI?*

We agree with the Reviewer that our original description of this procedure was not sufficiently clear. This has now been expanded as follows (new words in red): “For samples larger than the detector’s **horizontal** field of view, 4000 equally spaced projections were acquired through a 360° sample rotation **with the axis of rotation at the edge of the field-of-view, and projections**”.

between 180° and 360° “flipped” and joined to the 0°-180° ones to create a full dataset over 180°. Also in this case the exposure time per projection was of 0.2s, resulting in a total exposure time of approximately 800s. For samples larger than the detector’s vertical field of view, multiple scans were acquired at different vertical displacements of the sample, and the resulting volumes stitched together.”.

5. page 20, first line: *how did you train the ML model? did you label some slices (for one sample, or for each sample), or did you use readily trained model as such? It is not clear how to reproduce this step.*

Our apologies, we had taken this for granted as the training mechanism is described in the cited reference (former ref. 40, now 51) – we have now added the following sentence: “The same procedure described in the above reference was used to train the algorithm, consisting in manually segmenting a sub-set of the slices at regular intervals throughout the volume.”.

6. page 20, equations 1.1 - 1.3: *to me it seems that eq 1.1, 1.2 and 1.3 are not strictly necessary, as these calculations are rather trivial, and they are already well explained in the text.*

While overall we would agree with the Reviewer, also Reviewer 2 raised an important comment with regard to this (point 5), which we addressed through a modification of the equations themselves. It would look as if the modified equations may be helpful to prevent similar misunderstandings by the readers, and considering they should not excessively affect the manuscript’s readability as they only appear in the methods section, we would be inclined to leave them in; however, we remain of course open to any additional suggestions the Reviewer might have.

7. page 20, HB segmentation: *as the HBs are rather small in size, there is chance that choosing the segmentation somewhat differently would lead to quite different results. it would be good to indicate how reliable the HB volumes are.*

Please see response to point 2 (and to some extent point 1) by Reviewer 2, and especially the additional material we have added to the manuscript to address it. We hope this goes some way towards addressing this point, while of course acknowledging that some degree of uncertainty will inevitably remain.

Reviewers' comments:

Reviewer #2 (Remarks to the Author):

I would like to express my gratitude to the authors for addressing all of my critiques. Their responses have been satisfactory and have alleviated any concerns I had regarding the initial version of the manuscript. The manuscript has grown in length, which may impact readability to some extent, but it is undeniably clearer and more detailed.

I have a few minor comments:

1) On page 6, when discussing the comparison with histological images, it is mentioned that they are obtained from the same samples acquired with SPC-CT. Given this assertion, I am unsure why images from samples of different ages are shown instead of those from the same ages as the CT images.

2) In response to comment No. 5, the authors have clarified how volumetric contents were estimated. Considering the straightforward nature of this definition, I concur with Reviewer No. 3's suggestion to remove the equations and instead provide a concise definition, possibly moving the equations to supplementary materials.

3) Regarding the quantitative agreement between histology and SPC-CT, one may argue that histological sections are indeed representative of the entire volume. The authors have emphasized in various parts of the manuscript the added value of the proposed technique, highlighting that volumetric quantifications are more reliable than those based on a single or a few histological slices. I recommend adding an important piece of information from the caption of the supplementary table into the main text, specifically addressing the distortion and loss of tissues that may occur during histological preparation. Additionally, it is worth mentioning the artifacts introduced during cutting, which can significantly impact the interpretation of the tissue.

4) When considering the training of the algorithm used for image segmentation, I am curious about its reliability when applied to samples acquired with the laboratory system. Specifically, the question arises: Can a network trained on synchrotron-based images accurately segment your lab-based images? It would be beneficial if the authors could provide segmentation results comparing the same sample acquired with both modalities. This comparison has the potential to add value, particularly in the context of potential clinical applications with compact setups.

Reviewer #3 (Remarks to the Author):

The authors have done a good job updating the manuscript to clarify issues point out by me and the other reviewers, and they have also given satisfactory answers to those points that were not possible to ameliorate in the manuscript itself. I have no further critical comments to add.

Reviewer #4 (Remarks to the Author):

I think the authors' have addressed Reviewer #1's comments #1 and #3 adequately, but not comments #2 and #4.

I agree with Reviewer 1's comment #2, that putting the study in the context of the suggested references would help readers understand the advances represented by the manuscript under review. However, these studies were not cited or addressed in the revision, as far as I could tell. I understand that the authors feel that the references are not relevant, but I think discussing them would only strengthen the case that their study represents advances over published work.

I also agree with reviewer #1's concern regarding the DEC205 staining and comparison in fetal and postnatal thymus. I think the authors' figure included in the rebuttal supports the reviewer's concern, because the DEC205 stain does appear weaker in fetal (especially earlier stages) samples compared to postnatal samples. Moreover, the gating in the revised figure 4c and 4d comparing fetal and postnatal samples is problematic. The SSC/CD45 gates are set quite differently for fetal and postnatal samples (figure 4c/d), as are the Epcam/DEC205 gates. It appears to me that the SSC/CD45 gate set for the fetal samples is set around debris rather than cells, precluding analysis of epithelial populations.

We thank the Reviewers for their careful evaluation of our revised manuscript and their overall positive assessment. In the following, we provide a point-by-point response to their remaining comments. Changes to main manuscript and supplementary materials are highlighted in blue in the resubmitted version for ease of visualization.

REVIEWER 2

I would like to express my gratitude to the authors for addressing all of my critiques. Their responses have been satisfactory and have alleviated any concerns I had regarding the initial version of the manuscript. The manuscript has grown in length, which may impact readability to some extent, but it is undeniably clearer and more detailed.

Thanks for your appreciation of the effort we have put into revising the manuscript – we agree the revision resulting from incorporation of the Reviewers' comments has significantly improved it.

I have a few minor comments:

1) On page 6, when discussing the comparison with histological images, it is mentioned that they are obtained from the same samples acquired with SPC-CT. Given this assertion, I am unsure why images from samples of different ages are shown instead of those from the same ages as the CT images.

Images in figure 2 are extracted from exactly the same specimens, and matched. Conversely, images in figure 1b come from different specimens, and are simply meant to show that a similar evolution is observed with both SPC-CT and conventional histology. These H&E images were introduced in the (first) revision phase in response to a comment by one of the reviewers, at which point too much time had elapsed and it was impossible to perform H&E on the samples originally scanned at the synchrotron (the direct comparison of Fig 2, other hand, was included in the original submission). To clarify this, we have added the following sentence:

“Note that these examples come from different specimens and are simply meant to show that a similar evolution is observed with SPC-CT and more conventional approaches, with a more specific, direct comparison between the SPC-CT and H&E being presented below”

We thank the Reviewer for spotting this, and apologise if the original text was potentially confusing.

2) In response to comment No. 5, the authors have clarified how volumetric contents were estimated. Considering the straightforward nature of this definition, I concur with Reviewer No. 3's suggestion to remove the equations and instead provide a concise definition, possibly moving the equations to supplementary materials.

All equations have been moved to the supplementary materials.

3) Regarding the quantitative agreement between histology and SPC-CT, one may argue that histological sections are indeed representative of the entire volume. The authors have emphasized in various parts of the manuscript the added value of the proposed technique, highlighting that volumetric quantifications are more reliable than those based on a single or a few histological slices. I recommend adding an important piece of information from the caption of the supplementary table into the main text, specifically addressing the distortion and loss of tissues that may occur during histological preparation. Additionally, it is worth mentioning the artifacts introduced during cutting, which can significantly impact the interpretation of the tissue.

We thank the reviewer for this very good suggestion. The following sentence has been added to the discussion (“their use” refers to the SPC-CT datasets, as should be clear from the preceding sentence):

“An additional advantage associated with their use is the avoidance of problems caused by the distortion or loss of tissue that may occur during histological preparation, and of potential artifacts introduced during cutting, both of which can significantly impact the interpretation of the tissue”.

4) When considering the training of the algorithm used for image segmentation, I am curious about its reliability when applied to samples acquired with the laboratory system. Specifically, the question arises: Can a network trained on synchrotron-based images accurately segment your lab-based images? It would be beneficial if the authors could provide segmentation results comparing the same sample acquired with both modalities. This comparison has the potential to add value, particularly in the context of potential clinical applications with compact setups.

We have added a figure to the supplementary materials (Fig.S6) to show that an ML-based automated segmentation algorithm can effectively segment also the laboratory-based images. We note that in this case, the algorithm was trained on the laboratory image itself, which we believe better meets the Reviewer’s point about “potential clinical applications with compact setups” (as these would not have immediate access to synchrotron data). As well as potentially less powerful in terms of practical use, segmenting laboratory images using synchrotron data comes with additional challenges which we think lay beyond the scope of the present paper, also because, as explained in the manuscript, there were factual differences between the synchrotron and the lab images (“although the same sample was scanned, locating exactly the corresponding slice proved difficult, due to possible sample deformation occurring during transportation from the overseas synchrotron”).

It is also worth noting that ML methods have evolved significantly since our original data analysis, and freeware plugins such as WEKA (in Fiji-Image J) can now perform the same task as the algorithm described in our original reference 51, which we believe is beneficial to the wider community - especially on images obtained from lab scans. Please also note that one additional author (TP) was included as a result of the above additional task.

REVIEWER 3

The authors have done a good job updating the manuscript to clarify issues point out by me and the other reviewers, and they have also given satisfactory answers to those points that were not possible to ameliorate in the manuscript itself. I have no further critical comments to add.

Thank you very much for your positive and encouraging comment.

REVIEWER 4

I think the authors' have addressed Reviewer #1's comments #1 and #3 adequately, but not comments #2 and #4.

-I agree with Reviewer 1's comment #2, that putting the study in the context of the suggested references would help readers understand the advances represented by the manuscript under review. However, these studies were not cited or addressed in the revision, as far as I could tell. I understand that the authors feel that the references are not relevant, but I think discussing them would only strengthen the case that their study represents advances over published work.

We thank the Reviewer for the suggestion to include the references which may help the readers to further appreciate the advances in the field of the current manuscript. We have now added the recommended references into the main manuscript at page 4 and page 11 respectively:

“In the past, conventional CT has been employed to detect relevant macroscopic changes to diagnose thymic neoplasia and monitor overtime progression. However, the lower resolution and limits in soft tissue sensitivity of conventional CT allowed measuring the overall size, but not the anatomical compartmentalisation of the organ (Bogot and Quint, 2005; Liu et al, 2022; Dai et al, 2023)”

“Previously, conventional CT was employed to volumetrically quantify the total volume of murine thymic lobes, and match resulting values with histological projections. However, sub-compartmental characterisation was not achieved (Sakata et al., 2018).”

-I also agree with reviewer #1's concern regarding the DEC205 staining and comparison in fetal and postnatal thymus.

I think the authors' figure included in the rebuttal supports the reviewer's concern, because the DEC205 stain does appear weaker in fetal (especially earlier stages) samples compared to postnatal samples.

The “weaker” staining in foetal compared to postnatal samples at IHC is a visual problem due to the bright, high-density Hoechst+ nuclei - especially in the cortex compartment - and not to a lower number of cortical epithelial cells positive for CD205. We now provide the same figures with panels where CD205-positive cells (cortex) in

foetal samples are shown together with KRT14 (prevalent in medulla regions) without interference of nuclei staining so that CD205-expressing cells can be better appreciated. Please see Figure1 below.

Figure 1: Representative immunofluorescence images depicting cTEC (CD205, green) and mTEC (KRT14, red) across development. Nuclei were counterstained with Hoechst. Red blood cells autofluorescence, as observed in the perivascular space. CD205 positive cells and KRT negative in the medulla (white arrows) represent CD205 positive dendritic cells. KRT14 stains also the subcapsular layer in the cortex. Scale bar: 50µm.

Moreover, the gating in the revised figure 4c and 4d comparing fetal and postnatal samples is problematic. The SSC/CD45 gates are set quite differently for fetal and postnatal samples (figure 4c/d), as are the EpCAM/DEC205 gates. It appears to me that the SSC/CD45 gate set for the fetal samples is set around debris rather than cells, precluding analysis of epithelial populations.

The apparent gating problem is due to the compressed axis for CD45 shown in logarithmic scale which may create the effect of potential debris. CD45 gating is now shown with logarithmic scale not compressed in the new Figure 4c/d included in the revised manuscript, and the gating is clearly around a CD45-negative cell population. EpCAM and CD205 gating are set based on each corresponding tissue negative control (stained for live/dead only) that is now available in Supplementary Figure 4. We would like to point out that the data shown in the plots relative to the postnatal epithelial cells were acquired on a different instrument compared to those of the foetal one as the postnatal sample was processed for FACS sorting. Therefore, the analysis includes a much higher number of cells, and more events were recorded compared to the foetal tissue.

More importantly, the FACS plots of early-stage foetal samples may appear to contain proportionally less TECs than later stages. This is explained by the fact that we do not perform stromal cell enrichment as it would cause a considerable loss of cells leaving very few for the analysis. Early foetal samples are indeed much smaller and must be processed accordingly.

It is a matter of fact that thymic epithelial cells represent less than 0.1% of total thymic cellularity, which is composed mainly of CD45+ thymocytes. Therefore, to evaluate a sufficient number of events and calculate the ratio of cortical and medullary cells, it is necessary to remove as much as possible the CD45+ immune cells and enrich for stromal cells which contain TECs and other cell types (e.g., mesenchymal, endothelial etc). Please, refer to the Figure 2 below published in Ragazzini et al. 2023 describing the method used to enrich for stromal cells. In the current manuscript, enrichment was used for all samples GW17 or above but not at early stages GW11-16.

We have now clarified in figure legends and made it clearer in the method section which samples were processed for stroma enrichment and which ones were not. All these additions are highlighted in blue in the text of the main manuscript.

Figure2. Representative FACS plot showing stromal CD45- (left) and cTEC and mTEC populations (right) after several washes to remove thymocytes (left panels) and after stromal enrichment (right panels). Ragazzini et al., Dev Cell 2023.

REVIEWERS' COMMENTS:

Reviewer #2 (Remarks to the Author):

I thank the authors for responding to all my requests.

I have no further comments to make.

I believe that the manuscript can be accepted.

Reviewer #4 (Remarks to the Author):

I appreciate the effort spent addressing my comments. I have no further concerns.